

# Advancing multi-categorization and segmentation in brain tumors using novel efficient deep learning approaches

Nadenlla RajamohanReddy and G. Muneeswari

School of Computer Science and Engineering, VIT-AP University, Amaravati, Guntur, Andhra Pradesh, India

## ABSTRACT

**Background:** A brain tumor is the development of abnormal brain cells, some of which may progress to cancer. Early identification of illnesses and development of treatment plans improve patients' quality of life and life expectancy. Brain tumors are most commonly detected by magnetic resonance imaging (MRI) scans. The range of tumor sizes, shapes, and locations in the brain makes the existing approaches inadequate for accurate classification. Furthermore, using the current model takes a lot of time and yields results that are not as accurate. The primary goal of the suggested approach is to categorize whether a brain tumor is present, identify its type and divide the affected area into segments.

**Methods:** Therefore, this research introduced a novel efficient DL-based extension residual structure and adaptive channel attention mechanism (ERSACA-Net) to classify the brain tumor types as pituitary, glioma, meningioma and no tumor. Extracting features in brain tumor analysis helps in accurately characterizing tumor properties, which aids in precise diagnosis, treatment planning, and monitoring of disease progression. For this purpose, we utilized Enhanced Res2Net to extract the essential features. Using the Binary Chaotic Transient Search Optimization (BCTSO) Algorithm, the most pertinent features in terms of shape, texture, and colour are chosen to minimize complexity.

**Results:** Finally, a novel LWIFCM_CSA approach is introduced, which is the ensemble of Local-information weighted intuitionistic Fuzzy C-means clustering algorithm (LWIFCM) and Chameleon Swarm Algorithm (CSA). Conditional Tabular Generative Adversarial Network (CTGAN) is used to tackle class imbalance problems. While differentiating from existing approaches, the proposed approach gains a greater solution. This stable improvement in accuracy highlights the suggested classifier's strong performance and raises the possibility of more precise and trustworthy brain tumor classification. In addition, our method's processing time, which averaged 0.11 s, was significantly faster than that of previous approaches.

## INTRODUCTION

Brain tumors, the primary cause of brain cancer, are the most common type of brain disease. It is crucial to diagnose brain tumors early on in this type of cancer because it is

Corresponding author
G. Muneeswari,
muneeswari.g@vitap.ac.in

lethal. It is brought on by an unnatural and uncontrollably high brain cell count (*Kumar & Kumar, 2023*). There are numerous classification systems for brain tumors. Brain tumors are often classified as benign or malignant, one of the most popular classification schemes (*Balamurugan & Gnanamanoharan, 2023*; *Akter et al., 2024*). Brain tissue is not where benign brain tumors originate; instead, they form on the inside of the skull. Sometimes benign brain tumors can pose a severe threat to life. Approximately 85% of benign tumors are meningiomas. Meningiomas account for roughly 33% of cases and are slow-growing tumors. Women are diagnosed with meningiomas more often than men (*Deepa et al., 2023*; *Ullah et al., 2024*). Given their low propensity to spread to nearby brain tissue, they have a good chance of being surgically removed.

Even so, meningiomas can sometimes grow into cancerous tumors. The pituitary glands, which regulate hormones and physiological functions, are the site of origin of pituitary tumors. Non-cancerous pituitary tumors do not metastasize to other organs (*Nanda, Barik & Bakshi, 2023*). Pituitary tumors rarely result in cancer, but they can cause long-term hormone insufficiency, which can cause blindness. Malignant tumor cells are aberrant cells that proliferate erratically and uncontrollably (*Farajzadeh, Sadeghzadeh & Hashemzadeh, 2023*; *Ullah et al., 2022*; *Solanki et al., 2023*). These tumors can compress, invade, or destroy normal tissues.

Brain tumor detection techniques fall into two categories: DL-based and machine learning (ML)-based methods (*Aamir et al., 2023*; *Raza et al., 2022*; *Özkaya & Sağiroğlu, 2023*). Principal component analysis, fuzzy C-means, decision trees, support vector machines, and conditional random forests are examples of machine learning techniques that are frequently applied in diverse applications (*Shahin, Aly & Aly, 2023*; *Sobhaninia et al., 2023*). While DL is capable of learning and making decisions, ML bases its decisions on its prior knowledge (*Jabbar et al., 2023*). As a result, DL techniques are now widely applied. This approach, which is AI-based, allows for multi-level calculations (*Kumar & Kumar, 2023*; *Sharif et al., 2024*; *Allah, Sarhan & Elshennawy, 2023*). The classification of Deep Learning is based on supervised and unsupervised methods (*Kishanrao & Jondhale, 2023*), and DL demonstrated improved performance in many medical applications (*Saurav et al., 2023*; *Fernando & Tsokos, 2023*).

This study aims to create an automated model in the medical domain that is more accurate, improved, and effective in helping healthcare experts identify brain tumors at an early stage. This is because, when contrasted to conventional methods, it is becoming increasingly essential to overcome issues of time consumption, false observations, and limited expert accessibility. In comparison to the current state-of-the-art approaches, the proposed model also avoids imbalanced classes, overfitting, the need for high computational resources, and inadequate generalization on unseen data. It also exhibits improved, efficient, and more accurate results at early phases in the medical field while providing timely treatment to patients suffering from brain tumors.

Existing methods have difficulty in accurately classifying brain tumors due to variations in tumor size, shape, and location. Furthermore, many of these approaches are computationally expensive and time-consuming, resulting in diagnostic delays. Moreover,

datasets with class imbalances frequently produce biased predictions. Our proposed approach, which combines an efficient ERSACA-Net, Enhanced Res2Net for feature extraction, and novel optimization and segmentation techniques, specifically addresses these limitations by enhancing classification accuracy, reducing processing time, and efficiently dealing with class imbalance, making it more suitable for reliable brain tumor classification.

## Motivation

The classification and detection of brain tumors are motivated by the critical need for an early and precise diagnosis, which significantly improves medical outcomes and survival rates. While traditional methods rely heavily on manual analysis, which is time-consuming and prone to subjectivity, recent advances in AI and deep learning provide powerful tools for automating and improving brain tumor detection from medical images. However, current AI-based methods frequently struggle to balance accuracy, computational efficiency, and generalizability across diverse datasets.

To overcome these limitations, our proposed approach incorporates the ERSACA-Net approach, which distinguishes it from previous research. This method improves detection precision while minimizing computational cost, ensuring broader applicability in real-world medical settings. By incorporating this novel methodology, we hope to significantly improve the scalability, consistency, and accessibility of brain tumor detection systems, resulting in a more robust solution that better meets the needs of modern healthcare.

## Contribution

The major key contributions of this research are as follows:

- Leveraging publicly accessible datasets ensures a diverse and comprehensive collection of brain tumor images, enhancing the model's generalizability and robustness.
- Using CTGAN to tackle class imbalance ensures that the model is trained on a more balanced dataset, which mitigates bias and enhances the model's ability to classify less common tumor types accurately.
- Utilizing the Enhanced Res2Net method for multiscale feature extraction, capturing intricate details such as shape, texture, and colour is crucial for accurate tumor classification.
- By selecting important features and eliminating unnecessary ones, the Binary Chaotic Transient Search Optimization (BCTSO) algorithm reduces the complexity of the experimental performance, leading to faster computation and improved model efficiency.
- To categorize the tumor types a novel DL based ERSACA-Net is introduced.
- The ensemble of Local-information Weighted Intuitionistic Fuzzy C-means (LWIFCM) and Chameleon Swarm Algorithm (CSA) enhances clustering performance by combining local information weighting with an effective optimization algorithm, improving the precision and reliability of tumor detection.

**Article organization:** The remaining sections of the document are arranged as follows: A thorough analysis of all datasets and methodologies is provided in "Related Works", which also covers data preprocessing, classification and segmentation techniques. The results and the additional training and validation techniques are described in detail in "Proposed Methodology". Lastly, "Result and Discussions" contains an appendix with the conclusion.

## RELATED WORKS

This section thoroughly covers some relevant articles because of the substantial contributions that models based on deep learning can make to this field.

### Deep learning based classification and segmentation

The enhanced fully automatic segmentation (IFAS) convolutional neural network (CNN) model is suggested by *Kulshreshtha & Nagpal (2024)*. In IFAS, brain MRI images are segmented using a fully automatic algorithm and morphological operations. The CNN framework is employed for categorization and the U-net structure is considered for morphological segmentation in the assessment of the fully-automatic segmentation method.

To precisely identify and categorize tumor cells from MRI images using the Crossover Smell Agent Optimized Multilayer Perception (CSA-MLP) was introduced by *Arumugam et al. (2024)*. Preprocessing is done on the images to eliminate unwanted noise after they are gathered Brain tumor datasets. The images' features are extracted to carry out the process of categorization following preprocessing. It is also possible to classify healthy and unhealthy brain cells using the CNN classifier. To reduce errors and improve the efficacy of the suggested method, the multilayer perceptron (MLP) is utilized to categorize the category.

An enhanced deep learning-based framework for effective brain tumor detection is suggested by *Mandle, Sahu & Gupta (2024)*. Preprocessing techniques are used with the compound filter, which consists of the Gaussian, mean, and median filters, to enhance the quality of brain images. The texture and intensity patterns are extracted using the GLCM based method to identify tumor areas. Utilizing the Whale Social Spider-based Optimization Algorithm (WSSOA)-based metaheuristic, the best selection of features was carried out. Lastly, the deep convolutional neural network (DCNN) was employed to precisely detect tumors.

*Lee, Chae & Cho (2024)* presented an improved algorithm for computer-aided diagnosis tailored to classify brain tumors. Gaussian filters were used to eliminate noise from the MRI data, and Grid Mask was used to enhance the deep learning models' generalization capabilities. Next, utilize a technique they had developed to lessen the problem of brain tumors being hidden by regular grid masks: patterned grid masks.

Creating an algorithm that combines the principles of the Interval Type-II fuzzy logic system (IT2FLS) and artificial bee colony to identify the tumor region, which is surrounded by intricate brain tissues was introduced by *Alagarsamy, Govindaraj & Senthilkumar (2023)*. The key to any successful therapeutic sequence is the oncologists'

ability to make snap decisions. The algorithm described in this article dramatically enhances decision-making by utilizing technology.

*Asiri et al. (2023)* suggested an MRI-based brain tumor detection method that uses the SVM classifier and U-Net framework. The proposed study is predicated on enhancing and filtering MRI images to remove noise and improve contrast. To extract the area of interest, the modified U-Net architecture is utilized for MRI segmentation. The normal and tumor-affected images are classified using the support vector machine (SVM) algorithm following segmentation.

## Machine learning-based based classification and segmentation

Classification and detection of brain tumor using Kernel based SVM was proposed by *Rao & Karunakara (2022)*. The next step was to extract features using a combination of GLCM and SGLDM strategies. The Harris Hawks Optimization (HHO) algorithms were employed for selecting the features. Subsequently, KSVM-SSD was utilized to perform categorization. Here, the brain tumor was categorized as benign or malignant employing KSVM, and the malignant tumor was further categorized as low, medium, or high employing the social ski driver (SSD) optimization algorithm. *Sekhar et al. (2021)* developed a model based on TL to categorize the tumors into three types. The characteristics of the brain MRI images were extracted employing a pre-trained CNN, *i.e.*, Google LeNet. The characteristics are subsequently categorized using classifiers like softmax, SVM, and K-Nearest Neighbor.

*Amin et al. (2024)* suggested an unsupervised clustering strategy for tumor segmentation. Furthermore, a fused characteristic vector is employed, consisting of Gabor wavelet features (GWF), histograms of oriented gradient (HOG), local binary pattern (LBP), and segmentation-based fractal texture analysis (SFTA) characteristics. The random forest (RF) classifier was used to distinguish among three subtumoral regions: complete, enhancing, and non-enhancing tumor.

## Major challenges

Among the problems with standard BT classification models are the following:

Although machine learning has made significant progress in classifying and segmenting brain tumors, large research gaps still present opportunities for new deep learning-based methods. Managing tumor heterogeneity and the lack of high-quality annotated data are two obstacles. Integrating multimodal imaging data and creating interpretable, broadly applicable models are still challenging. In particular, when it comes to small or asymmetrical tumors, current models frequently have trouble correctly drawing the borders of the tumors. There are still issues with seamlessly integrating AI models into clinical workflows, prospectively validating them in real-world scenarios, and connecting imaging features to genomic data. It's also crucial to manage the high computational demands of advanced models, address model biases, and ensure patient data privacy. By utilizing cutting-edge methods for data augmentation, multimodal learning, interpretability, and efficient computation, novel deep-learning approaches can close these gaps and create reliable, accurate, and clinically valuable tools for brain tumor analysis.

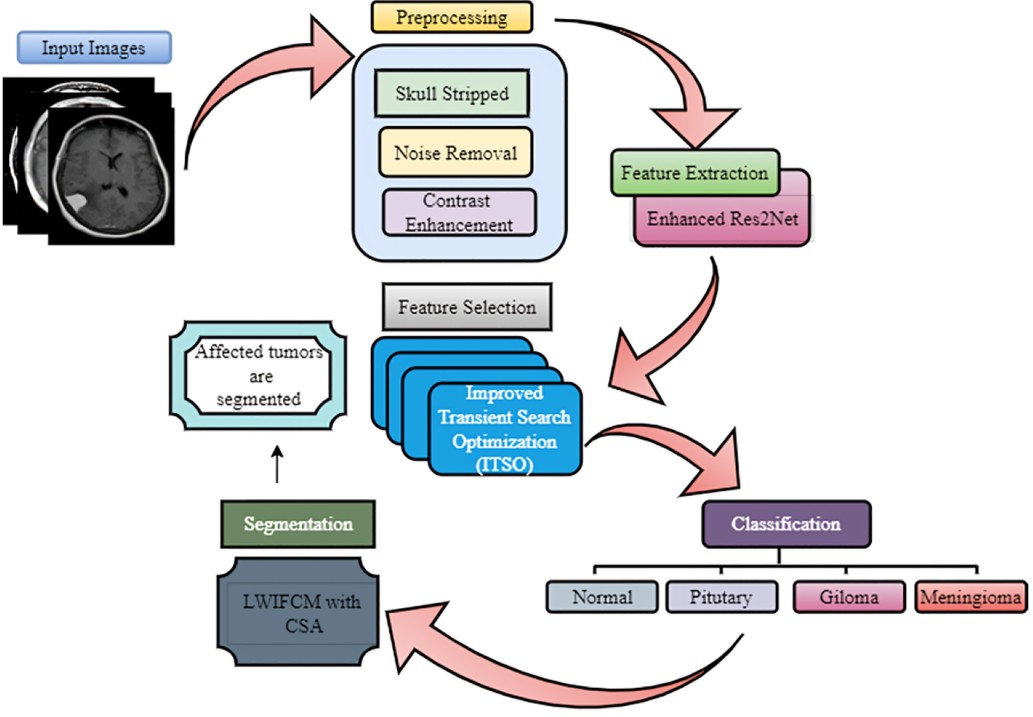

**Figure 1 Overall framework of the proposed methodology.**

# PROPOSED METHODOLOGY

Early BT detection is essential for treatment planning and patient care. For BT to be manually classified using MRI with similar structures or appearances, the radiologist's skill and experience in identifying and classifying BT is necessary. The main objective is to create and refine an efficient method. Initially, the raw data is fed into the preprocessing phase. The raw is noise-reduced, skull-stripped and enhanced. Then, the significant features are extracted and selected with the help of the Binary Chaotic Transient Search Optimization (BCTSO) Algorithm and Enhanced Res2Net.

We introduced a novel DL-based ERSACA-Net to categorize the tumor types. Then, to segment the tumor categories, we utilized an efficient ensemble Local-information weighted intuitionistic Fuzzy C-means clustering algorithm (LWIFCM) and Chameleon Swarm Algorithm (CSA). Figure 1 shows the overall proposed framework.

## Preprocessing

The following are the main actions that will be taken on the MRI images in this step to ensure that the system can read the correct input and create a better environment for image analysis:

**Skull removal:** Since the focus of neuroimaging analysis is on brain tissue, the skull and surrounding structures are removed. Advanced image processing techniques are employed to accurately segment and extract the brain tissue from MRI scans, eliminating irrelevant

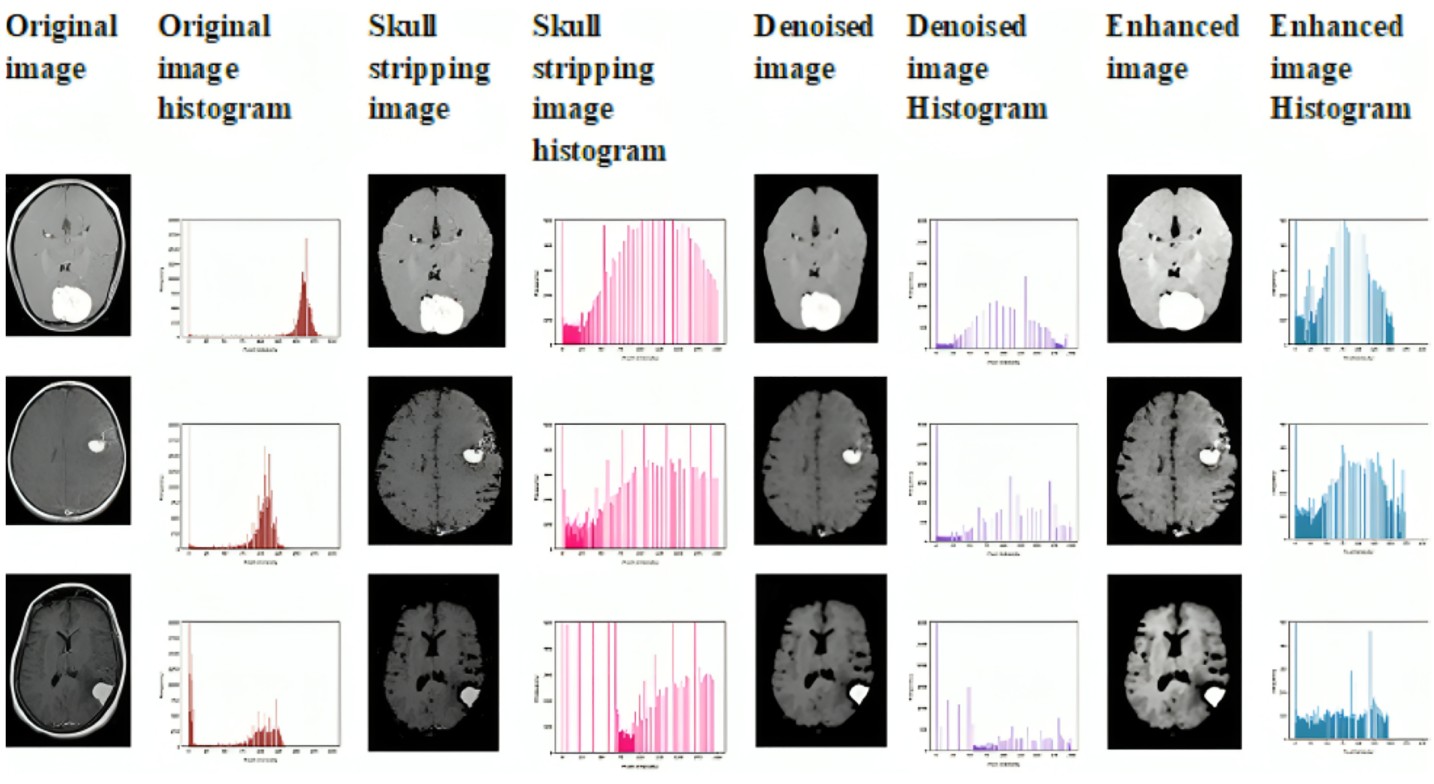

**Figure 2 Sample preprocessed image with their histogram.**               

background noise and artifacts. This process ensures that subsequent analysis focuses solely on the brain, improving the precision of classification and segmentation tasks.

**Image filtration:** A median filter is applied to the MRI images to reduce noise while preserving essential features. This filter is particularly effective in enhancing the quality of MRI images by suppressing unwanted noise and highlighting important anatomical structures. As a result, this step ensures cleaner input data, which improves the reliability of the machine-learning algorithms used for brain tumor analysis.

**Contrast enhancement:** To further improve image quality, we utilize Adaptively Clipped Contrast Limited Adaptive Histogram Equalization (ACCLAHE). This technique enhances local contrast by adaptively adjusting the contrast limits for different regions of the image. By doing so, ACCLAHE significantly increases the visibility of tumor boundaries and tissue differentiation, making the features more distinguishable for accurate classification and segmentation. This contrast enhancement is crucial for improving the overall performance of the model, especially in detecting fine details in brain tumor images. The sample preprocessed image is shown in Fig. 2.

## Data augmentation

To address the imbalanced data problem and lessen network fitting, CTGAN-based data augmentation is carried out following the preprocessing. It subtracts a few samples from majority classes and adds more samples to minority classes. It is an architecture based on

GANs and intended for tabular data synthesis. Reducing the difficulties associated with using GAN structure for tabular data modelling is the main objective of CTGAN's features. More precisely, the CTGAN architecture handles non-Gaussian and multimodal distribution by using the mode-specific normalization (MSN) that transforms constants of any distribution into a bounded vector. This is an appropriate representation for neural networks.

The two neural network stages of CTGAN are a generator G and a critic C, which resembles the discriminator in a traditional GAN architecture. To manage the non-Gaussian and multimodal variation of successive fields in tabular data, CTGAN employs mode-specific normalization. An approach to the issue of imbalanced collections in continuous columns is to use sample-based training in conjunction with a conditional generator. Additionally, CTGAN uses some of the most recent developments in GAN training, such as the loss function of WGAN-GP and the critical framework of PacGAN, to improve the quality of generated data and training stability. The following equation displays the CTGAN loss function.

$$L = E_{G(z) \sim P_g}[D(G(z)) - E_{x \sim P_r}[D(x)] + \lambda E_{y \sim P_y}\left[(\nabla_y D(y) - 1)^2\right]. \tag{1}$$

Here, the gradient penalty factor is represented by the symbol $\lambda$, and the sample y is continuously interpolated to the actual data x. Pr and Pg show how the generated and accurate data are distributed. The generator in CTGAN receives conditioning information (y) and random noise (z) from the tabular data (T). In addition to other characteristics of the final data, the conditioning information may define the range or type of samples in each column. After that, the discriminator (D) receives the fake samples the generator produced to reach a final judgment. Once the discriminator has established *via* backpropagation if the instance is genuine or fraudulent, it sends a sign to the generator. The generator uses this indication to adjust its weights and enhance its capacity to generate accurate data. Figures 3 and 4 show the distribution of categories after and before augmentation.

## Feature extraction

The process of feature extraction is essential to categorization. To accurately represent images of brain tumors, we retrieve the colors, textures, and shapes that are essential in them. It is difficult to extract the best features from brain images. The feature extraction process converts unprocessed data into numerical data while preserving the original content. Features can be extracted using manual models or automated ones. While all significant features are extracted by manual feature extraction, only issue-related significant features are extracted by automated feature extraction. We extracted the features using the Enhanced Res2Net model.

For brain tumor analysis, the Enhanced Res2Net approach to feature extraction provides superior multiscale representation, enhancing the accuracy and robustness of tasks related to classification and segmentation. With little computational overhead, it effectively captures both fine and coarse details. More accurate treatment planning and improved diagnostic support result from this.

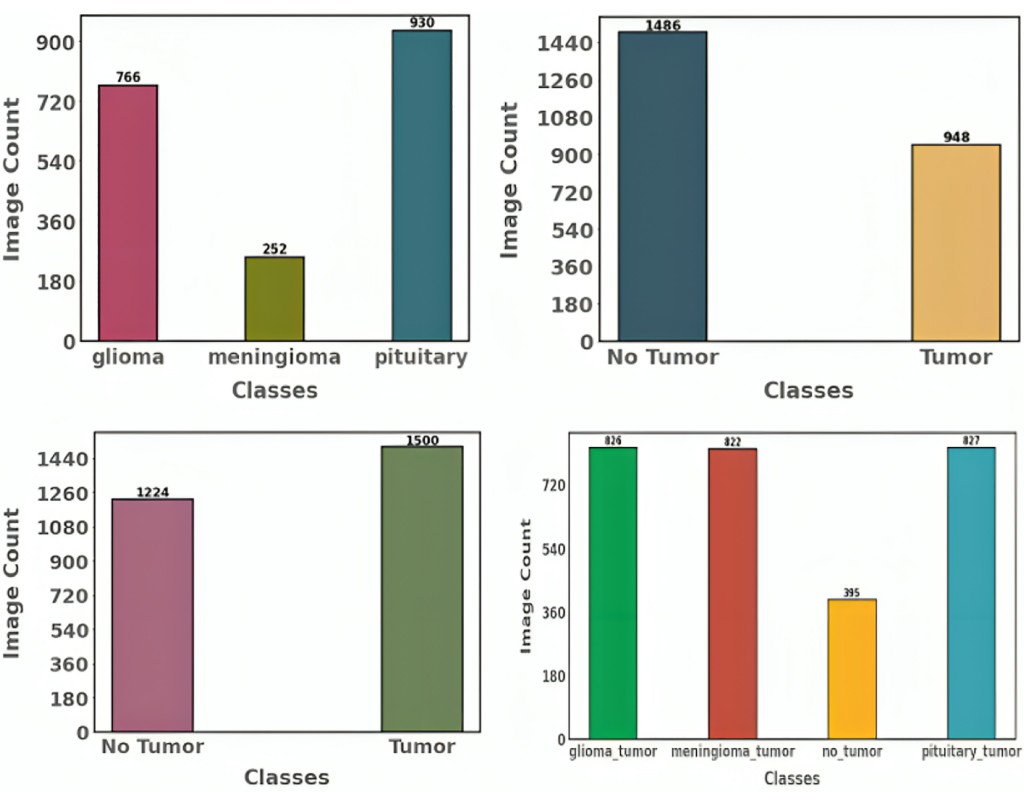

**Figure 3 Label distribution of proposed dataset before data augmentation.**

However, in the context of brain tumor categorization, standard deep learning models can struggle to detect the intricate details of MRI images, such as subtle differences in tumor shape, texture, and intensity, which are critical for accurate diagnosis. To capture multi-scale features and rich contextual information, we used an improved feature extraction approach called Enhanced Res2Net. This allowed us to focus on the most important aspects of the images, improving the classification process's accuracy and robustness.

While deep and transfer learning algorithms are well-known for their ability to extract characteristics from images, we chose to use Enhanced Res2Net for extracting features in our approach to address the unique challenges of brain tumor classification. Brain tumors vary significantly in size, shape, texture, and location, making it difficult for generic deep-learning models to capture all critical features effectively. Using Enhanced Res2Net, we can extract multi-scale features that provide a more complete and detailed representation of tumor characteristics. This multi-scale feature extraction captures not only the tumor's overall structure but also the fine details required for accurate classification.

## Res2Net approach

Res2Net performs better in terms of generalization than ResNet. The residual structure of cells of the model incorporates hierarchical small residual blocks, thereby augmenting the effective sensory field of every layer and enhancing the overall network's extraction of

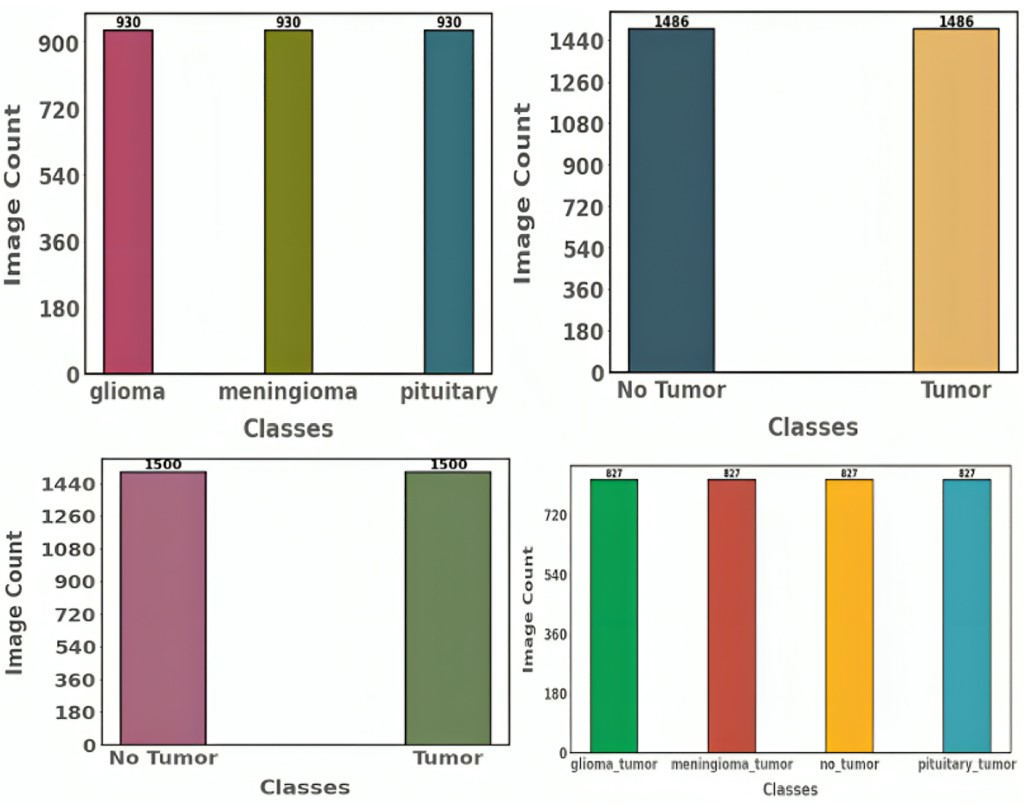

**Figure 4** Label distribution of proposed dataset after data augmentation.

features effectiveness. We chose the Res2Net model as the foundation network for the antler slice categorization task because of its ability to extract meaningful features from such images, considering the small size and difficulty of identifying deer antler slices.

## Enhanced Res2Net approach

The feature extraction module employed the Res2Net convolutional neural network as its foundational network. The three-by-three convolution group of the multi-level residual architecture replaced the bottleneck structure's three-by-three convolution. This expands the network's receptive field, allowing it to gather various degrees of fine-grained scale characteristic data regarding objects. This multiscale refers to the conjunction of various receptive fields at a finer granularity rather than the combination of levels. Initially, the input map of features underwent a 1 × 1 convolution process. Then, they were split equally into s-map subsets in the channel dimension. The number of channels in these subsets was decreased to 1/s of the input channels, but they still have the same scale size. The receptive field is then increased, and all the characteristic data collected in the previous stage is contained in every 3 × 3 convolution operation. While the hierarchical residual connections within distinct residual blocks can capture fine-grained changes globally and for details, Res2Net can obtain combined data on features of different numbers and receptive field sizes. To obtain the last feature information, the output characteristics of

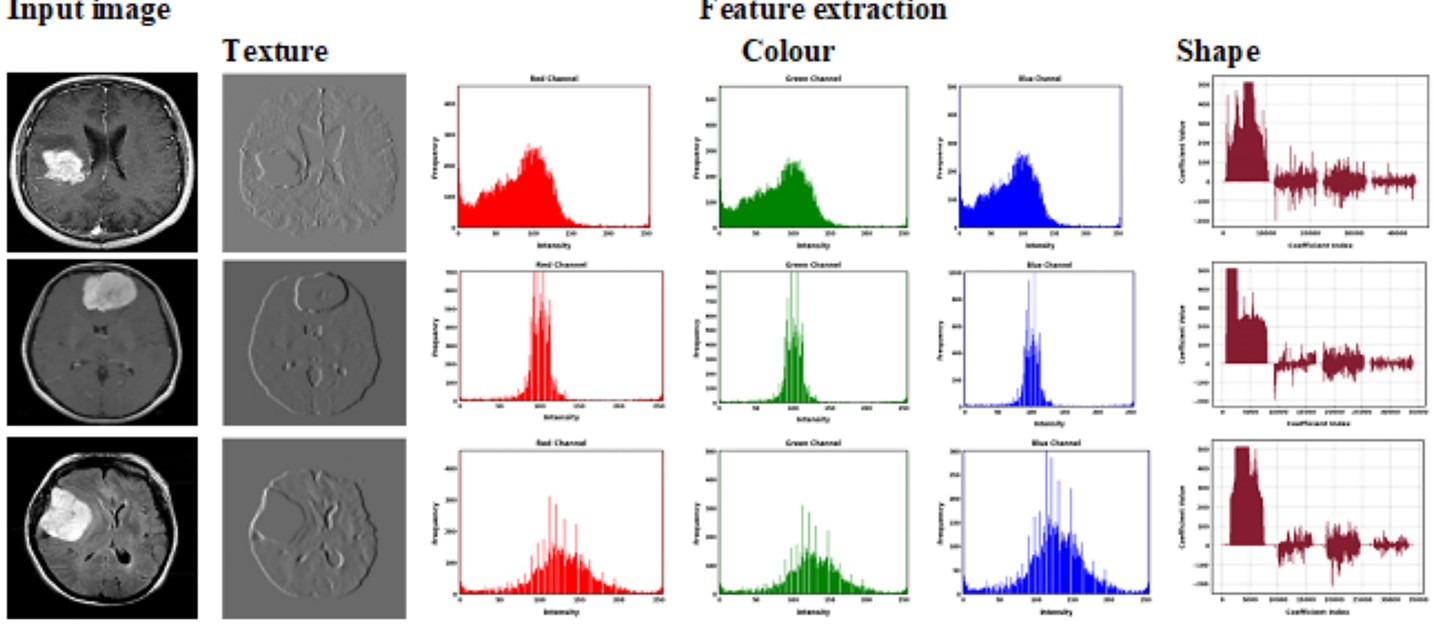

**Figure 5 Histogram equalization of extracted features.**

every phase were parallelized using the concat function. The extracted features histogram is shown in Fig. 5.

## Feature selection

In optimization problems, feature selection algorithms built on bio-inspired metaheuristic have proven successful and significant. Numerous wrapping-based algorithms that employ binary metaheuristic algorithms have been used to solve the selecting features issue. High-accuracy feature-obtaining algorithms are essential. We employed a Binary Chaotic Transient Optimization Algorithm (BCTSO) to select the features.

There are several advantages to using the BCTSO for brain tumor analysis feature selection, including improved accuracy, robustness, efficiency, and interpretability. This strategy improves model performance and guarantees scalability and adaptability to large-scale and complex medical imaging datasets by concentrating on the most pertinent features. In the end, this helps to improve the accuracy and dependability of brain tumor diagnosis and treatment planning.

## Chaos theory

Chaotic systems unpredictably behave in deterministic systems because of their seemingly random and irregular motions. Chaotic systems are defined by their inherent, ubiquitous nature. Chaotic behaviour is not completely disordered; instead, it is deterministic. Research has revealed that their initial values strongly influence the behaviour of chaotic systems. Chaotic models are additionally stochastic, ergodic, and sensitive to initial values.

## Transient search optimization

To optimize outcomes of searches based on the transient behaviours of switched electrical systems containing capacitors and inductances as storage elements, transient search optimization (TSO) is utilized. Equation (2) illustrates the transient and steady-state responses (final responses) of a circuit with fully reacting resistive components (R) and energy-storing components like capacitors (C), inductors (L), or both (LC).

$$Complete\ response = Transient\ response + Final\ response \tag{2}$$

A circuit with just one storing component (either RL or RC) is referred to as first-order. There are two storage components (RLCs) in a second-order circuit. No amount of switching these circuits will quickly shift the procedure toward a subsequent steady state. This allows one to compute the first-order circuit's transient response using Eq. (3).

$$\frac{d}{dt}x(t) + \frac{x(t)}{\tau} = K \tag{3}$$

The solution $x(t)$ can be found by solving the equation for differentials, as shown in Eq. (4).

$$x(t) = x(\infty) + (x(0) - x(\infty))e^{\frac{-t}{\tau}} \tag{4}$$

An RL circuit needs an inductor current (i(t)) to function. For an RL circuit, it is L/R, and for an RC circuit, it is RC. $\tau$ is the circuit's time constant. The response value at the end is indicated by $x(\infty)$. The transient effect of the second-order circuit can be found by solving the Formula (5).

$$\frac{d^2}{dt^2}x(t) + 2\alpha\frac{d}{dt}x(t) + \omega_0^2 x(t) = f(t) \tag{5}$$

Equation (6) illustrates how to get the solution to the second-order differential equation.

$$x(t) = e^{-\alpha t}(B_1 \cos(2\pi f_d t)) + x(\infty) \tag{6}$$

The damping coefficient, denoted by $\alpha$, the resonant frequency, denoted by $\omega_0^2$ the damped resonant frequency, and the constants $B_1 B_2$ are all included in this formula. There are three primary steps in TSO:

(1) Assign the search agents' upper and lower boundaries for the search area; (2) Consider options (3) Determining the best course of action.

$$Y = lb + rand \times (ub - lb) \tag{7}$$

TSO is exploitable since the initial order discharge decays exponentially. The random number, r1, is chosen so that the exploitation and exploration of TSO algorithms are balanced ($r_1 \geq 0.5$). The TSO algorithm is utilized and mathematically investigated, as shown in Eq. (7), obtained from Eqs. (5) and (7).

$$Y_{l+1} = \begin{cases} Y_l^* + \left(Y_l - C_l \cdot Y_l^*\right)e^{-T} & r_1 < 0.5 \\ Y_l^* + e^{-T}[\cos(2\pi T) + \sin(2\pi T)]Y_1 - C_l \cdot Y_l^* & r_1 \geq 0.5 \end{cases}. \tag{8}$$

The optimal solution $Y_l^*$ of the TSO algorithm approximates the state of equilibrium $x(\infty)$ of an electrical circuit

$$T = 2 \times z \times r_2 - z \tag{9}$$

$$C_1 = k \times z \times r_2 + l \tag{10}$$

$$z = 2 - 2(l/L_{max}()). \tag{11}$$

A search area's lower and upper bounds are denoted by *lb* and *ub*, accordingly. Equation (6) involves a uniformly distributed random number, rand, and a change in z from 2 to 0. T and $C_1$ are the random coefficients, and the random numbers $r_1$, $r_2$, and $r_3$ are uniformly distributed among 0 and 1. The position of a search agent is denoted by $Y_l$ optimal position by $Y_l^*$, and iteration number by l. $k$ is a constant and $L_{max}$ has the greatest number of iterations. Additionally, the process's exploration and exploitation balance is determined by the coefficient T, which varies between $(-2, 2)$. While the TSO algorithm is explored when T < 0, it is exploited when T > 0.

## Population encoding

Each individual was encoded using binary encoding. In this method, an individual is represented as a string of binary '0s' and '1s'. The feature is said to be enabled when its value is 1, and disabled when its value is 0. The sigmoid function was used to convert continuous numbers produced by ITSO operators to binary values, as demonstrated in Eq. (13).

$$sigmoid\left(x_i^d\right) = \frac{1}{1 + exp\left(-x_i^d\right)} \tag{12}$$

$$binary\left(x_i^d\right) = \begin{cases} 1 \; if \; sigmoid\left(x_i^d\right) > r_d \\ 0 \; else \end{cases}. \tag{13}$$

The range of the random number, $r_d$, is 0 to 1. T Using our dataset, we first apply cross-validation, choosing one fold for testing and the other four for training. During the training phase, the TSO algorithm generates a feature subset. The fitness function then assesses the quality of the generated subset of features. This iterative process illustrates the systematic approach used to train and evaluate our model since it is repeated until a preset amount of iterations is reached.

## Objective function

The suggested ITSO algorithm removes the superfluous attributes before choosing the best set of features. The algorithm uses fitness functions to complete this task. The fitness function unifies two competing goals: selecting features ratio $n/N$ and classifier effectiveness *Acc*. Fitness is calculated for every individual vector $Y_l$ using the Eq. (14).

$$Fit = w_1 \times Acc + w_2 \times \frac{n}{N}. \tag{14}$$

$N$ is the overall number of features, n is the number of characteristics that have been chosen, and $Acc$ denotes the classification accuracy. Accordingly, the weight coefficients allocated to the accuracy and selecting features ratio components in Eq. (14) are $w_1$ and $w_2$, respectively. The algorithm emphasizes accuracy while taking the significance of minimizing the set of features into account, as shown by the values of $w_1$ and $w_2$, which are set at 0.6 and 0.4.

### Improved transient search optimization (ITSO)

This section introduces an Improved transient search optimization (ITSO) algorithm. Without a doubt, the TSO method provides a quasi-optimal answer to an optimization issue. Chaotic maps were utilized in TSO to improve results and speed up convergence. The exploration space might not be thoroughly examined because most intelligent algorithms employ a random initialization population. Chaos theory has proven useful in many areas of mathematics, including algorithmic initialization.

Nonlinear deterministic bounded systems lack periodicity and convergence and are called chaotic systems. A chaotic system also depends significantly on its initial conditions and parameters. Because chaos is unpredictable, ergodic, regular, and random, it offers trustworthy randomness. Compared to random sequences produced from uniform distributions, chaotic sequences offer heuristic optimization algorithms a more efficient search strategy. In this article, a logistic map is employed to create solution sequences.

$$x_{t+1} = r \times x_t \times (1 - x_t), \ t = 1, 2, 3, \ldots, t_{max}. \tag{15}$$

A random coefficient slightly impacts the effectiveness of TSO. This problem might cause local optima to become stuck. You can go past this bottleneck by thoroughly exploring the search space. Because chaotic sequences are non-periodic, they have lower repetition values than conventional random data, which allows them to cover the entire search area. Our investigations show that chaotic maps produce this feature, and the TSO algorithm's coefficient C1 aids in searching the search space.

### Classification

Our suggested approach and its variations are thoroughly explained in this section. There are various advantages to categorizing brain tumors using the ERSACA-Net. By fusing adaptive attention with residual learning, ERSACA-Net improves feature extraction by directing the model's attention to the most pertinent channels and regions within the brain tumor images. This methodology enhances the precision of differentiating between tumor types by efficiently capturing minute details and tumor morphological variations. Furthermore, the network's capacity to adaptively weigh features guarantees strong performance in various imaging scenarios and patient demographics, which eventually contributes to more accurate and dependable tumor classification and improved diagnosis and individualized treatment planning.

Firstly, the CER-Block is introduced, combining channel expansion concepts and residuals to extract image information accurately. The ER-Net, the network's backbone for the categorization of brain tumor diseases, is built based on the CER-Block. Secondly, the design of the ACA-Block directs the network's backbone to concentrate on tumor disease data to minimize redundant data interference.

## CER-Block and ER-Net

Convolutional operations are typically used by traditional image classification networks to scale their channels, which can increase the number of parameters. Gradient information will disappear as the network gets deeper because many training settings lead to a significant computational aspects load. By triple-expanding the amount of channels without adding more parameters this aids in broadening the perceptual field. Subsequently, the data aggregation layer receives the features acquired through max-pooling, enabling the network to concentrate on leaf disease data from various angles.

CER-Block is the foundation upon which ER-Net, the leading network, is built. The CER-Net comprises three CER-Blocks and two downsampling layers, as illustrated in Fig. 6. The CER-Block consists of a residual link layer and data extracted from the characteristics of images. Three maximum pooling layers with varying window sizes and a data aggregation layer comprise the image characteristic data extraction layer. A basic convolutional layer finds it challenging to convert composite characteristic data from simple to abstract because tumor diseases frequently appear as combinations of attributes like color, texture, and shape.

## ACA-Block and ERC-Net

The CBAM integrates spatial and channel attention. Let $x$ the kernel's size in the convolution and $y$ be the feature map's channel count. Next, the following represents the original Gaussian likelihood density function:

$$y = \frac{1}{\sigma\sqrt{2\pi}} e^{-\frac{(x-\mu)^2}{2\sigma^2}}. \tag{16}$$

The following is a comparison of the formulas used to determine the map of the featured data:

$$F'_{BAM} = BN(MLP(Avgpool(F))) + M_s(F) \tag{17}$$
$$F'_{CBAM} = \sigma(MLP(Avgpool(F)) + MLP(Maxpool(F))) + M_s(F) \tag{18}$$
$$F'_{ACA-Block} = F + \sigma(IGPDF(Avgpool(F) + IGPDF(Maxpool(F))) + M_s(F) \tag{19}$$

where the outcome of the CBAM, BAM, and ACA-Block are, in that order $F'_{BAM}$, $F'_{CBAM}$, and $F'_{ACA-Block}$. Batch-normalization is known as BN. Two two-dimensional convolutions combine to form the multilayer perceptron known as MLP. SPACE is represented by $M_s(F)$. F is the input data. Inverse Gaussian likelihood density function, or IGPDF for short. Average pooling is known as AvgPool. MaxPool is the maximum pooling. To increase the accuracy of the classification of brain tumors, as depicted in Fig. 7, the

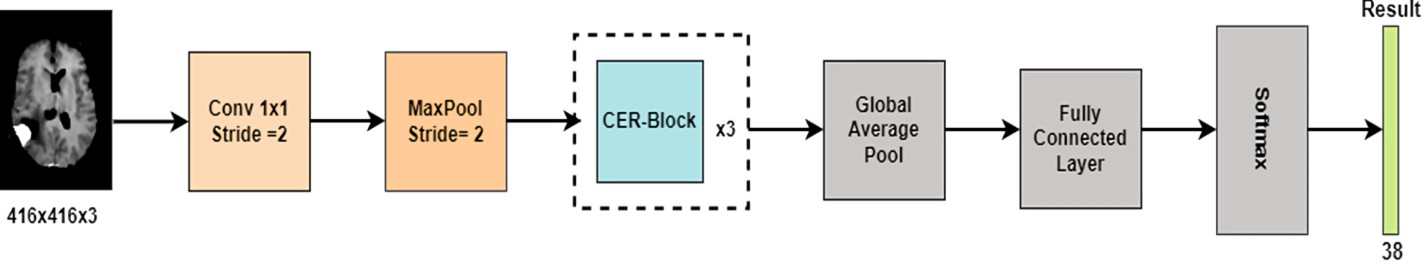

**Figure 6 ER-Net network structure.**

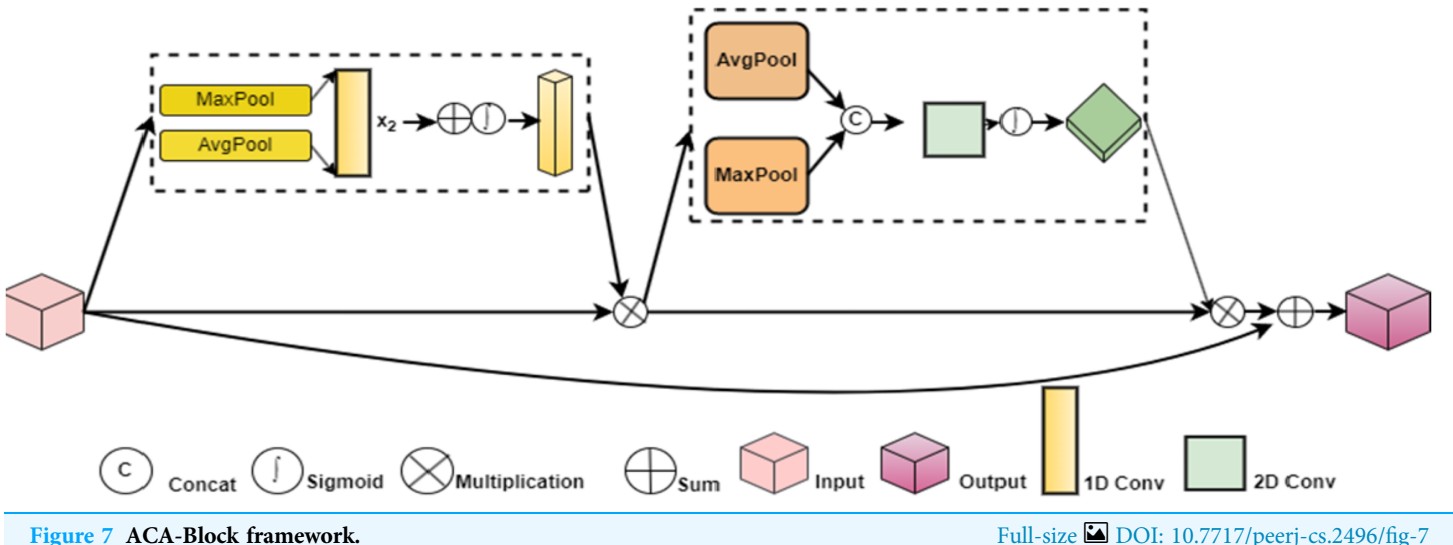

**Figure 7 ACA-Block framework.**

ACA-Block can focus on tumor features, remove unnecessary data, and further boost the information connections among the feature maps.

## ERCP-Net

The output layer of the conventional image categorization network receives the prediction results. However, the semantic information obtained from the network's lower layers is all that can be sent to this output layer. It is challenging to concentrate on the image's pixel-level details in the interim. The enhanced output layer can produce a reliable prediction result by concentrating on semantic and pixel-level data. The ERCP-Net network's structure is shown in Fig. 8.

The semantics is applied to the shallow data, and more profound information about pixels is added, resulting in a combined map of features with more channels. Further semantic and pixel information refinement is achieved by downsampling and recombining the merged feature data with the characteristics data from the last CER-Block+ACA-Block structure. Feedback from the final set of information is sent to the output layer for accurate

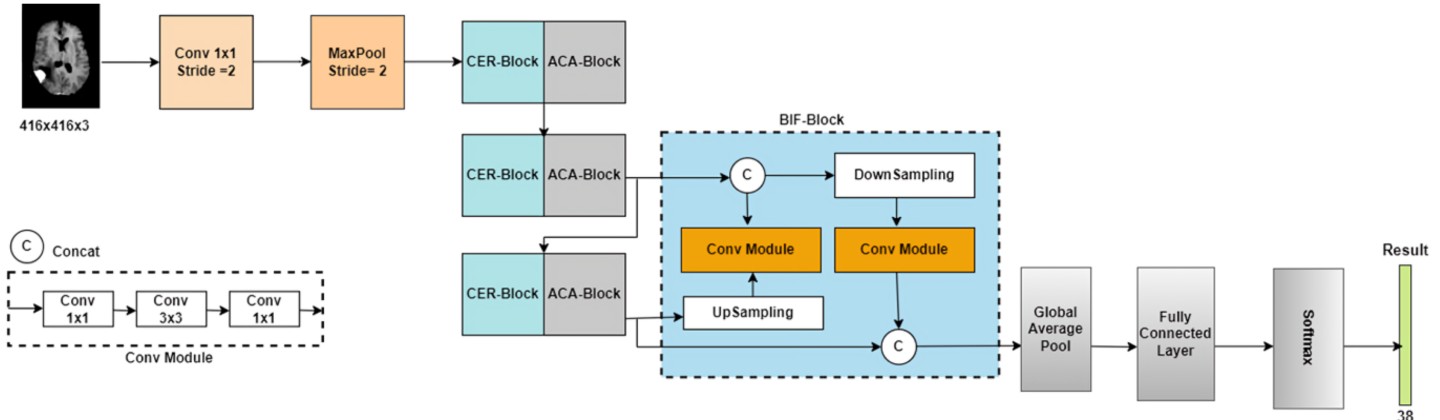

**Figure 8 The framework of the ERCP-Net network.**

**Table 1 The ERCP-Net specifications and the tensor sizes of every result layer.**

| Layer | Size of the tensor | Layer | Size of the tensor |
|---|---|---|---|
| Input | (3, 416, 416) | CER_Block_3 | (432, 13, 13) |
| Conv | (16, 208, 208) | ACA_Block_3 | (432, 13, 13) |
| MaxPool | (16, 104, 104) | BIF_Block | (1,296, 13, 13) |
| CER_Block_1 | (48, 52, 52) | Global average pool | (1,296, 1, 1) |
| ACA_Block_1 | (48, 52, 52) | Fully connected layer | (1,296) |
| CER_Block_2 | (144, 26, 26) | Softmax | (38) |
| ACA_Block_2 | (144, 26, 26) | | |

classification results. Table 1 shows all of the output tensor dimensions of the ERCP-Net layers.

## Segmentation based on LWIFCM_CSA approach

We introduced LWIFCM_CSA to segment the tumors. By combining the advantages of both methods, the ensemble of the Chameleon Swarm Algorithm (CSA) and the LWIFCM approach provides improved segmentation accuracy. While CSA maximizes cluster centres for better boundary delineation, LWIFCM efficiently manages image noise and intensity inhomogeneity. In complex medical imaging scenarios such as brain tumor delineation, this fusion approach improves the robustness and precision of segmentation tasks.

## Intuitionistic FCM algorithm

By including two additional parameters, the degree of non-membership $\gamma$ and the degree of uncertainty are denoted as $\pi$ the intuitionistic fuzzy sets (IFS), a significant expansion of fuzzy sets, delicately capture the fuzzy nature of the objective world. On the set $X$, the IFS $A$ is specified as

$$A = \{u_A(x), \gamma_A(x), \pi_A(x) | \forall x \in X\}. \tag{20}$$

In the case when $(x) \rightarrow [0, 1]$, where $0 \leq u(x) + \gamma A(x) \leq 1$. Furthermore, $(x) = 1 - (x) - \gamma(x)$ can be used to define the degree of uncertainty. The following is a definition of intuitionistic fuzzy entropy (IFE):

$$IFE(A) = \sum_{i=1}^{n} \pi_A(x_i) exp(1 - \pi_A(x_i)). \tag{21}$$

The degree of non-membership is written as

$$\gamma_A(x_i) = (1 - u_A(x_i)^{\alpha})^{1/\alpha}, x \in X. \tag{22}$$

While the uncertainty parameter is called $\alpha$. The degree of uncertainty can then be stated as

$$\pi_A(x_i) = 1 - u_A(x_i) - (1 - u_A(x_i)^{\alpha})^{1/\alpha}, x \in X. \tag{23}$$

Consequently, the IF set $A$ is written as follows:

$$A^{IFS} = \left[u_A(x_i), (1 - u_A(x_i)^{\alpha})^{1/\alpha}, 1 - u_A(x_i) - (1 - u_A(x_i)^{\alpha})^{1/\alpha} | x \in X\right]. \tag{24}$$

Moreover, the fuzzy membership degree in the original FCM is changed to the intuitionistic fuzzy membership degree, which forms the IFCM clustering algorithm. The IFCM's objective function, $J^*(U, V)$ is described as

$$J^*(U, V) = \sum_{i}^{c} \sum_{j}^{n} u_{i,j}^{*m} d^2(x_j, v_i) + \sum_{i=1}^{c} \pi_i^* exp(1 - \pi_i^*). \tag{25}$$

The membership degree $U$ updating mode in the IFCM remains unchanged, while the clustering centre $V$ modifying mode is modified to

$$v_i^* = \left(\sum_{j=1}^{N} u_{i,j}^{*m} x_k\right) / \sum_{j=1}^{N} u_{i,j}^{*m}. \tag{26}$$

Nevertheless, there are some disadvantages to the IFCM clustering algorithm, including its sensitivity and noise susceptibility.

## LWIFCM approach

This section proposes the LWIFCM clustering algorithm to solve the aforementioned algorithms' drawbacks. To avoid over-reliance on local data in low-noise regions and undervaluing the impact of the degree of membership, the local weight of information is denoted as $k_{ij}$ is incorporated to modify the extent of the adaptive influence of local information on clustering outcomes. Conversely, the suggested clustering algorithm can fully utilize the local information factor $G$ in high noise regions. The local information

weight $k_{ij}$ and the optimization function with the objective in the LWIFCM are represented as

$$\hat{J}(U, V) = \sum_{i}^{c} \sum_{j}^{n} \hat{u}_{ij} d^2(x_j, v_i) + \sum_{i=1}^{c} \pi_i^* exp(1 - \pi_i^*) + k_{ij} G_{ij} \tag{27}$$

$$k_{ij} = \frac{\sigma_j^2 + \rho}{\overline{\sigma}^2 + \rho} \tag{28}$$

whereas $\overline{\sigma}^2$ is represented as mean squared error of the sample, and $\sigma_j^2$ is the variance. The terms $Nj$ and $k$ have definitions that align with Eq. (28).

## Boosted LWIFCM approach with Chameleon Swarm algorithm

Two factors primarily influence the segmentation approach based on fuzzy clustering: the initial clustering centers selection and the key parameter setting. The uncertainty parameter ($\alpha$) and fuzzifier constant ($m$) are essential parameters in the LWIFCM and are typically found through trial and error investigations. The choice of starting clustering centers significantly impacts the clustering outcomes, which are correlated with the accuracy of image segmentation. To maximize the effectiveness of clustering techniques and boost adaptability, algorithms based on swarm intelligence are typically added. This article proposes the Chameleon Swarm algorithm (CSA), a novel swarm intelligence algorithm that mimics the social behaviours of a swarm and is inspired by those behaviours. This section introduces the CSA implementation and the clustering algorithm based on the CSA (CSA-LWIFCM).

The CSA initializes the population as a metaheuristic algorithm to facilitate the optimization process. Assume that there are C people in the population overall and they are in the D search space. The way the initial population is produced in the dimension and the search space is randomly initialized can be expressed as follows:

$$a^i = L_j + rand \times (U_j - L_j). \tag{29}$$

The representation of an $i$ represents the $i^{th}$ chameleon's initial vector. In the $j^{th}$ dimension, the search area's lower and upper bounds are denoted, respectively, as $L_j$ and $U_j$. The number generated at random and falling between 0 and 1 is called a rand. The improved chameleons' search capabilities in the field of search can be stated as follows:

$$\rho = \delta \, exp^{(-\alpha t/R)}. \tag{30}$$

In this case, $\rho$ is the iteration-specific parameter that decreases as the number of iterations increases. The preset parameters that are used to control the exploration and exploitation phases are $\delta$, $\alpha$, and $\beta$. The following are the revolving centred coordinates that are used to update the chameleons' positions in the search space:

$$arand_r^i = m \times ac_r^i. \tag{31}$$

The chameleon's rotating centred coordinates are called $rand_r^i$. The rotation matrix is represented by m, and the centring coordinates at the r$^{th}$ iteration are represented by $ac_r^i$. The iteration inertia weights are provided as follows:

$$W = (1 - r/R)^{\left(\lambda\sqrt{r/R}\right)}. \tag{32}$$

In this case, $W$ it represents the inertia's weight, and $\lambda$ is the random number regulating exploitation capacity. A value of one is associated with $\lambda$. The chameleon's rate of acceleration is expressed as

$$y = 2590 \times (1 - exp^{-\log(r)}) \tag{33}$$

where the chameleon's acceleration is calculated $y$. We see that the CSA starts the optimization process and uses the equations to update the locations of the chameleons.

$$a_{r+1}^{i,j} = \begin{cases} a_r^{i,j} + p_1\left(P_r^{i,j} - G_r^j\right)rand_1 + p_2\left(G_r^j - a_r^{i,j}\right)rand_2 & rand_i \geq P \\ a_r^{ij} + \rho(U^j - L^j)rand_3 + L_b^j\text{sgn}(rand - 0.5) & rand_i < P \end{cases} \tag{34}$$

$$a_{r+1}^i = arand_r^i + a_r^{-1} \tag{35}$$

$$a_{r+1}^i = a_r^{-i} + \left(\left(v_r^{ij}\right)^2 - \left(v_{r-1}^{ij}\right)^2\right)/(2y). \tag{36}$$

$G_r^j$ It represents the chameleon's ideal global location and $v_r^{ij}$ denotes its new velocity. If a chameleon leaves the search space, it will return to its previously established constraints. The fitness function is estimated in each iteration to ascertain which chameleon is the fit. The fitness function is employed to determine which chameleon is the best at catching its prey first. These can be carried out repeatedly until the entire iteration cycle is satisfied.

### Research question

**R1**: How can the proposed method effectively classify and segment brain tumors from MRI scans, given the variability in tumor size, shape, and location?

**R2:** What advantages does the ERSACA-Net offer over existing deep learning-based classification approaches?

**R3:** How does the utilization of Enhanced Res2Net for feature extraction and Binary Chaotic Transient Search Optimization (BCTSO) for feature selection impact the accuracy and efficiency of brain tumor classification?

**R4:** Can the novel LWIFCM_CSA approach and CTGAN effectively handle class imbalance and improve segmentation performance compared to traditional methods?

**R5:** What improvements in classification accuracy and processing time can the proposed approach achieve compared to state-of-the-art techniques?

## RESULT AND DISCUSSIONS

In this section, the performance and efficacy of the suggested methods and present the results of our brain tumor categorization and segmentation approach in this nt metrics like recall, accuracy, precision, and the Dice similarity coefficient are used to assess the results.

We address the effects of our results and draw comparisons between our findings and current methods to identify areas for improvement. We also discuss possible drawbacks, the robustness of our techniques, and future research directions to improve brain tumor analysis even more.

A variety of hyperparameters were used in the training of our proposed models. The suggested models were trained with the following parameters: categorized cross-entropy loss function, Adam optimizer, 200 epochs, 32 batch size, and 0.0001 learning rate. The Softmax classifier was used for pretraining; however, scratch-based DL models have been proposed. A total of 80% of the data is used to train the suggested models, with the remaining 20% being used for validation and testing. The evaluation metrics for both estimated and ground truth labels have been computed for the brain tumor dataset.

## Experimental setup

The experiments were conducted on a system equipped with an Intel i5 2.60 GHz processor and 32 GB of RAM, running on Windows 10. The software environment utilized includes Python, Keras, and TensorFlow, all executed within the Anaconda3 framework.

## Dataset description

The datasets used in our experiments were chosen for their ability to represent the various challenges of brain tumor classification. These datasets cover a wide range of tumor types (pituitary, Glioma, meningioma, and no tumor), as well as characteristics like shape, texture, and location. To effectively test our CTGAN-based solution, we prioritized high-quality MRI scans for their clinical relevance and detailed imaging capabilities, as well as datasets with class imbalance. This ensures that our approach is reliable and applicable in real-world situations.

Brain MRI dataset: The dataset used to model the performance of the suggested approach is from the Kaggle license CCO: Public Domain. In total, there are 3,264 MRIs. The four classes of MRIs in the training dataset are represented by the numbers 826, 822, 395, and 827, respectively, for brain MRIs with gliomas, meningiomas, and pituitary tumors. The dataset was accessed from https://www.kaggle.com/datasets/masoudnickparvar/brain-tumor-mri-dataset.

BRATS 2020 dataset: The primary goals of the BRATS 2020 dataset (19) are to examine, evaluate, and disseminate high-quality data. Kaggle Datasets allows for the private or public sharing of datasets. The patient's chances of survival are the main focus. The training, validation, and testing data consists of numerous GBM/HGG and lower grade glioma (LGG), with pathologically confirmed diagnosis and available OS. Brain MRI. The dataset was accessed from https://www.kaggle.com/datasets/awsaf49/brats2020-training-data.

Figshare: We used 3,064 brain MRI slices from 233 patients from a public brain tumor data set from Figshare. It involves three different views (sagittal, axial, and coronal) and three different types of brain tumors (glioma, pituitary, and meningioma). The dataset was accessed from https://www.kaggle.com/datasets/rahimanshu/figshare-brain-tumor-classification.

Br35H dataset: The Brain Tumor Detection 2020 (BR35H) dataset is utilized, comprising 255 MRIs showing positive and 255 negative brain tumors. Both T1-weighted and T2-weighted image sequences are included in the dataset. The dataset was accessed from https://www.kaggle.com/datasets/ahmedhamada0/brain-tumor-detection.

## Performance metrics

The performance metrics assessed for the suggested method are accuracy, precision, recall, and F-score.

$$Accuracy = \frac{T_n + T_p}{T_n + F_n + T_p + F_p} \tag{37}$$

$$Precision = \frac{T_p}{F_p + T_p} \tag{38}$$

$$Recall = \frac{T_p}{F_n + T_p} \tag{39}$$

$$F - Score = 2 \times \frac{Precision \times Sensitivity}{Precision + Sensitivity} \tag{40}$$

$$MSE = \frac{1}{n} \sum_{i=1}^{n} ((a_i - a_i)^2 \tag{41}$$

$$RMSE = \sqrt{\frac{1}{n} \sum_{i=1}^{n} (a_i - \hat{a}_i)^2} \tag{42}$$

$$MAE = \frac{1}{n} \sum_{i=1}^{n} (a_i - \hat{a}_i) \tag{43}$$

## Analysis of brain MRI dataset (dataset 1)

To show enhancements in categorized brain tumors and segmentation, we thoroughly analyse the Brain MRI dataset in this subsection. We assess the effectiveness of our proposed methods on a number of metrics and compare the outcomes with those of previous approaches. Figure 9 shows the outcome of dataset 1.

The existing approaches like DNN, Adaptive ANFIS, 2DCNN and CNN-GA are utilized to compare with the proposed approach. Table 2 illustrates the differentiation of various approaches using dataset 1.

While differentiating from existing approaches, the proposed approach yields a greater solution, which is shown in Fig. 10.

## Analysis of Figshare dataset (dataset 2)

This subsection showcases the effectiveness and efficacy of our suggested techniques on the extensive Figshare dataset, providing a detailed analysis of our brain tumor segmentation and classification methods. The sample outcome of dataset 2 is shown in Fig. 11.

Table 3 and Fig. 12 show prior differentiation with the proposed approach using dataset 2. The table highlights the accuracy, precision, and recall of the different methods used for classifying brain tumors on the Br35H MRI dataset. The suggested approach

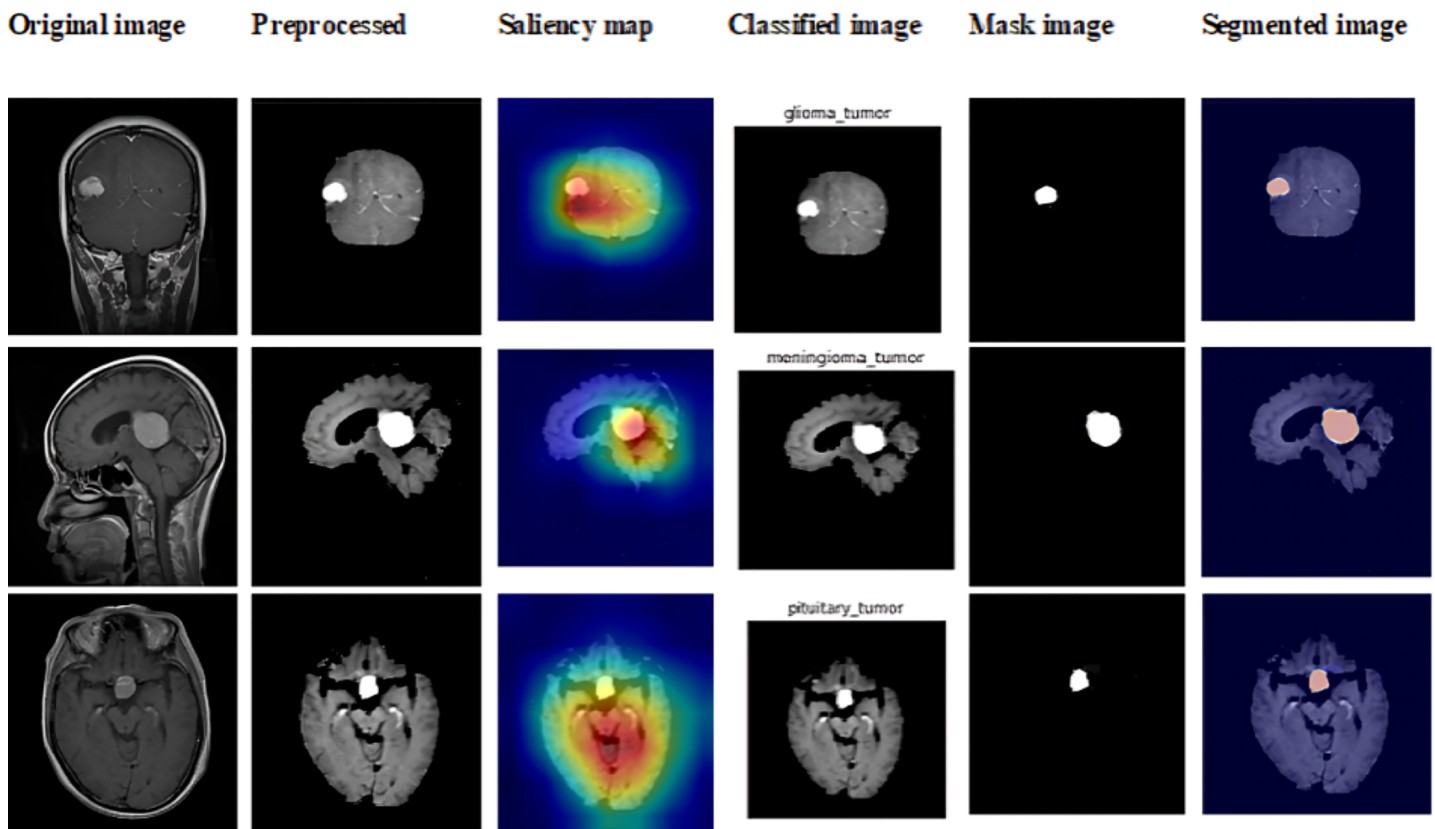

| Original image | Preprocessed | Saliency map | Classified image | Mask image | Segmented image |

**Figure 9** Sample outputs of dataset 1.

**Table 2** Differentiation of various approaches using dataset 1.

| Approach | Class | Accuracy (%) | Precision (%) | Recall (%) |
|---|---|---|---|---|
| DNN | No tumor | 96.46 | 95.45 | 95.46 |
| | Glioma | 96.46 | 95.63 | 95.74 |
| | Meningioma | 96.73 | 95.36 | 95.47 |
| | Pituitary | 96.13 | 95.63 | 95.37 |
| Adaptive ANFIS | No tumor | 97.35 | 98.46 | 98.68 |
| | Glioma | 97.15 | 98.73 | 98.74 |
| | Meningioma | 97.36 | 98.27 | 98.37 |
| | Pituitary | 97.37 | 98.21 | 98.62 |
| 2DCNN | No tumor | 96.46 | 97.57 | 98.54 |
| | Glioma | 96.75 | 97.14 | 98.36 |
| | Meningioma | 96.84 | 97.47 | 98.27 |
| | Pituitary | 96.85 | 97.78 | 98.73 |
| CNN based approaches | No tumor | 93.53 | 94.36 | 95.63 |
| | Glioma | 93.14 | 94.62 | 95.93 |
| | Meningioma | 93.62 | 94.52 | 94.63 |
| | Pituitary | 93.52 | 94.26 | 95.62 |

(Continued)

| Approach | Class | Accuracy (%) | Precision (%) | Recall (%) |
|---|---|---|---|---|
| CNN-GA | No tumor | 97.03 | 98.17 | 98.67 |
| | Glioma | 97.19 | 98.31 | 98.91 |
| | Meningioma | 97.25 | 98.25 | 98.89 |
| | Pituitary | 97.26 | 98.37 | 98.59 |
| Fuzzy C-Means | No tumor | 99.37 | 99.84 | 99.57 |
| | Glioma | 99.24 | 99.74 | 99.85 |
| | Meningioma | 99.50 | 99.85 | 99.84 |
| | Pituitary | 99.56 | 99.57 | 97.85 |
| Proposed | No tumor | 99.42 | 99.86 | 99.63 |
| | Glioma | 99.36 | 99.80 | 99.85 |
| | Meningioma | 99.57 | 99.89 | 99.88 |
| | Pituitary | 99.63 | 99.65 | 99.19 |

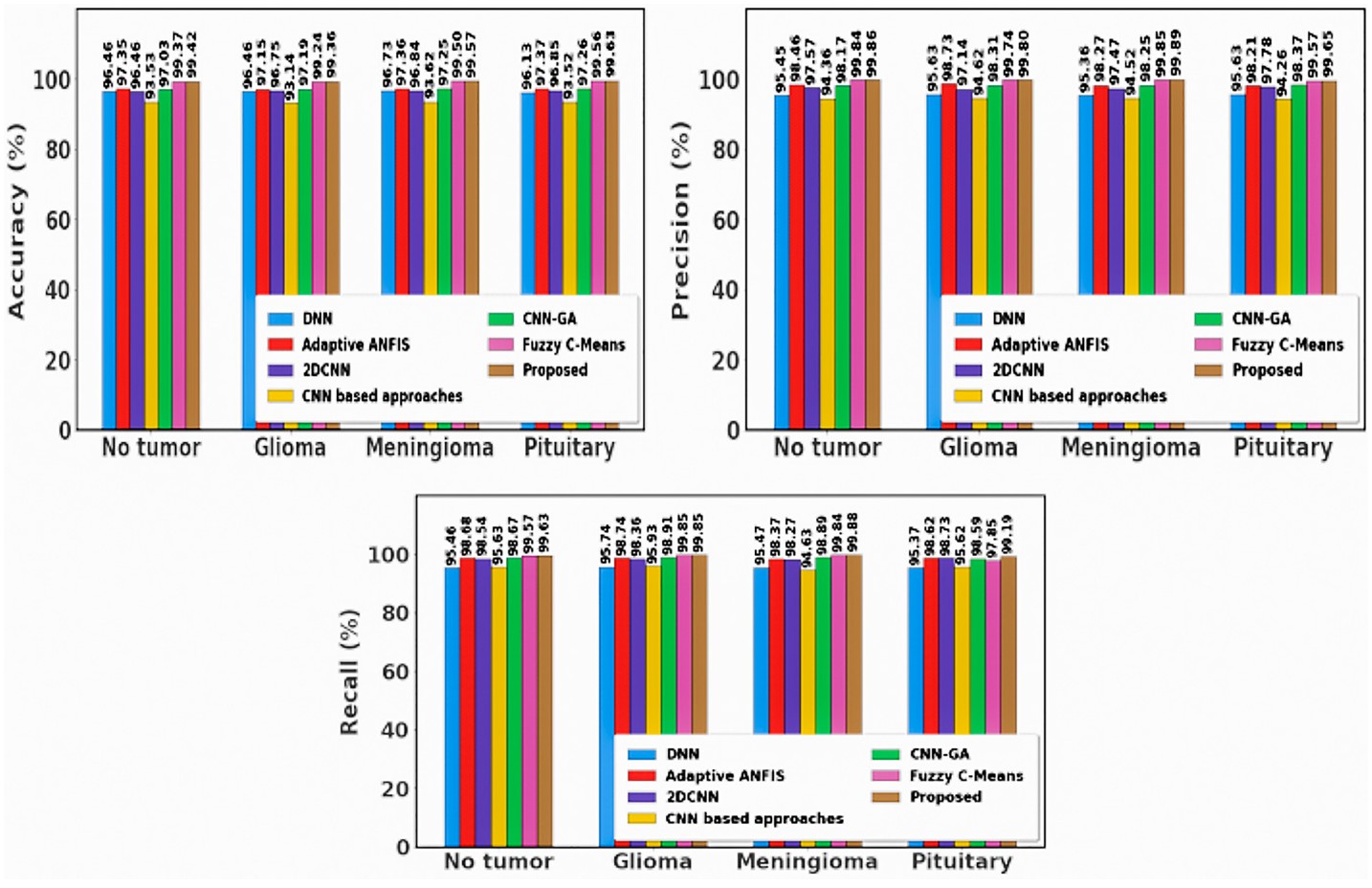

**Figure 10 Differentiation of prior with proposed approach using dataset 1.**

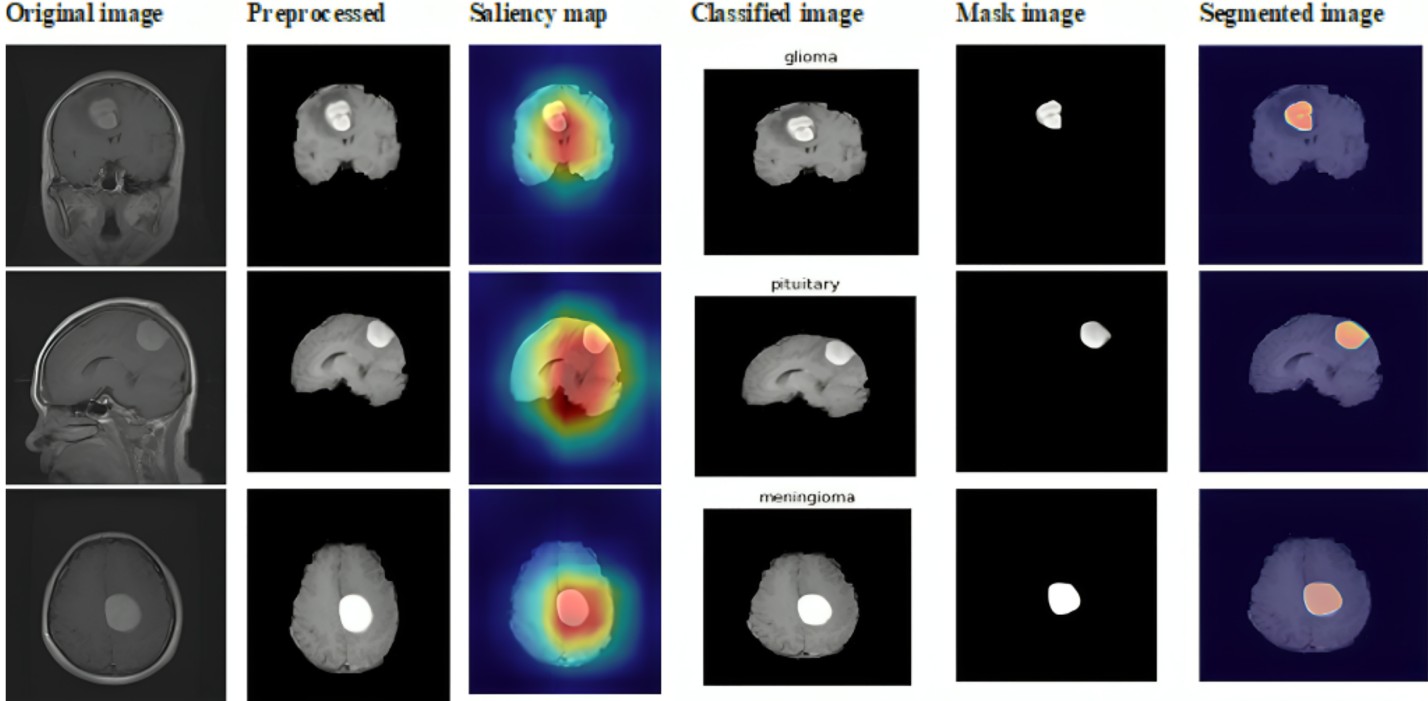

**Figure 11 Sample outputs of dataset 2.**

**Table 3 Differentiation of various approaches using the Figshare dataset.**

| Method | Class | Accuracy (%) | Precision (%) | Recall (%) |
|---|---|---|---|---|
| CNN | No tumor | 97.86 | 98.49 | 98.33 |
| | Glioma | 97.31 | 98.17 | 99.14 |
| | Meningioma | 97.59 | 97.33 | 98.64 |
| | Pituitary | 97.45 | 98.51 | 99.15 |
| DBFS-EC | No tumor | 98.31 | 97.22 | 89.74 |
| | Glioma | 94.86 | 98.49 | 98.30 |
| | Meningioma | 97.09 | 98.62 | 99.16 |
| | Pituitary | 98.09 | 94.59 | 94.59 |
| CNN-Bayesian | No tumor | 94.57 | 98 | 98.50 |
| | Glioma | 95.85 | 92.10 | 89.74 |
| | Meningioma | 95.45 | 98.24 | 94.91 |
| | Pituitary | 96.79 | 98.61 | 98.89 |
| KNN-SVM | No tumor | 94.89 | 95.90 | 94.59 |
| | Glioma | 95.04 | 97.51 | 98.98 |
| | Meningioma | 94.89 | 91.89 | 87.17 |
| | Pituitary | 97.17 | 96.49 | 93.22 |
| ADRU-SCM | Pituitary | 93.53 | 97.45 | 91.42 |
| | No tumor | 92.45 | 95.35 | 90.35 |
| | Glioma | 95.35 | 88.45 | 96.56 |
| | Meningioma | 95.45 | 90.35 | 97.45 |

(Continued)

| Table 3 (continued) | | | | |
|---|---|---|---|---|
| Method | Class | Accuracy (%) | Precision (%) | Recall (%) |
| PDCNN | Pituitary | 97.34 | 98.02 | 95.67 |
| | No tumor | 97.57 | 98.11 | 95.78 |
| | Glioma | 97.56 | 97.13 | 96.84 |
| | Meningioma | 97.37 | 97.24 | 97.47 |
| DCNN-GAP | Pituitary | 98.35 | 97.36 | 97.25 |
| | No tumor | 98.64 | 97.46 | 97.19 |
| | Glioma | 98.50 | 98.14 | 99.01 |
| | Meningioma | 98.36 | 97.43 | 97.43 |
| Fuzzy C-Means | No tumor | 98.56 | 99.14 | 99.25 |
| | Glioma | 98.24 | 98.66 | 99.19 |
| | Meningioma | 98.50 | 99.14 | 99.10 |
| | Pituitary | 98.56 | 97.43 | 97.43 |
| Proposed | No tumor | 99.08 | 99.25 | 99.32 |
| | Glioma | 99.13 | 99.11 | 99.24 |
| | Meningioma | 99.21 | 99.08 | 99.12 |
| | Pituitary | 99.09 | 99.10 | 99.05 |

performs better than the others, attaining the highest accuracy (99.13–99.21%) and consistently high precision and recall for every class of tumor. This illustrates how much better the suggested method is at recognizing and classifying different types of tumors.

## Analysis of BraTs dataset (dataset 3)

This subsection explores the analysis we performed on the BraTs 2020 dataset, which is a standard for brain tumor segmentation, to assess our suggested techniques. Using this complex and varied dataset, we evaluate the robustness and effectiveness of our models and offer comprehensive findings and insights. Figure 13 shows the sample outcome of dataset 3.

The performance of various methods for classifying brain tumors is compared in the table. With the highest accuracy (99.13%), precision (99.05%), recall (99.01%), and F-Score (99.03%), the proposed approach performs superior to any other technique. Remarkably, it outperforms the deep neural network (DNN), Differential Evolution Neural Network (DENN), Multi-SVM, artificial neural network (ANN), and others in terms of brain tumor classification. Figure 14 and Table 4 shows the comparison of dataset 3.

## Analysis on Br35H MRI dataset (dataset 4)

A thorough examination of the Br35H MRI dataset is provided in this subsection, with the primary objective being to assess how well our suggested techniques for brain tumor. The sample output is shown in Fig. 15.

The proposed approach achieves the best accuracy (99.20%), precision (99.02%), and recall (99.08%) compared to all other methods. Xception, MobileNetV2, InceptionV3, and

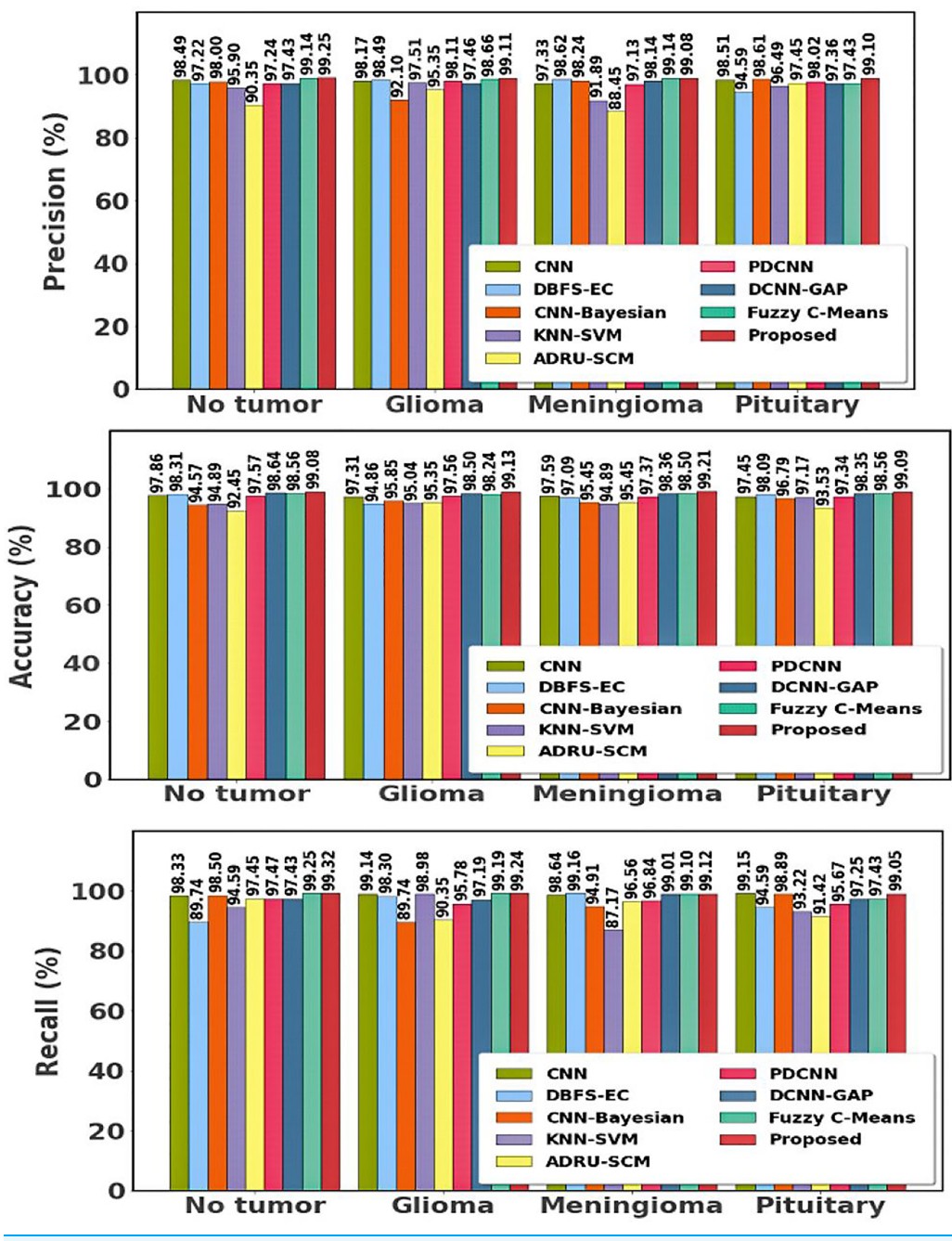

**Figure 12** Differentiation of prior with proposed approach using dataset 2.

EfficientB0, on the other hand, perform worse; EfficientB0 has the lowest accuracy (90.88%), precision (93.12%), and recall (89.12%). Figure 16 and Table 5 shows the differentiation of various approaches using dataset 4.

## Overall investigation over the proposed approach

In this section, we analyze the overall categorization and segmentation of the performances. Table 6 shows the performance impact of image enhancement approaches.
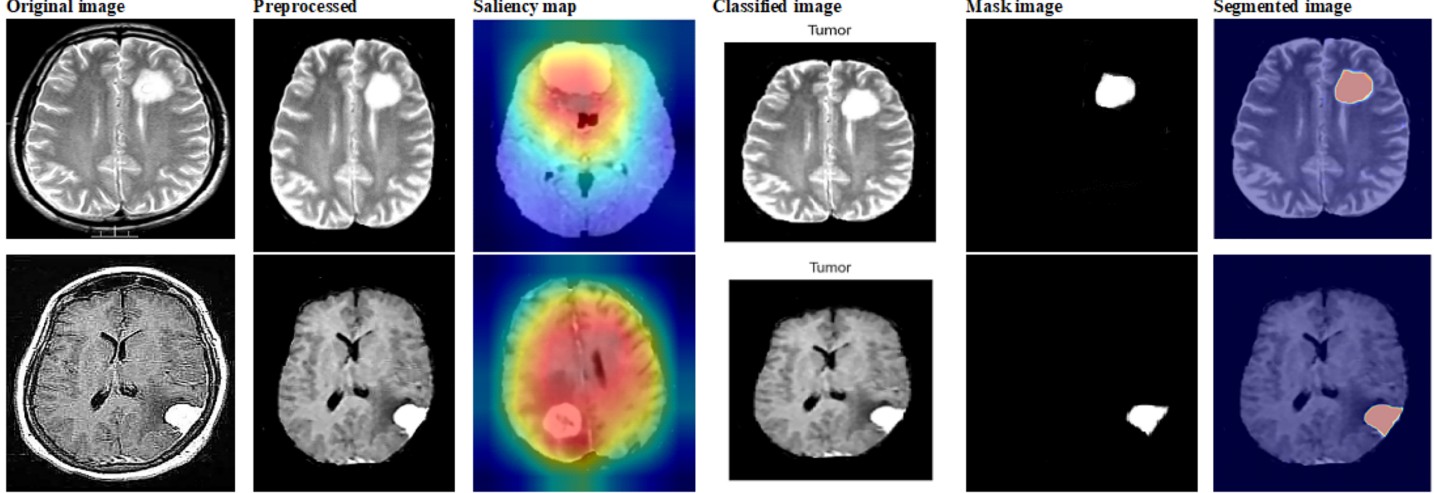

**Figure 13 Sample outputs of dataset 3.**

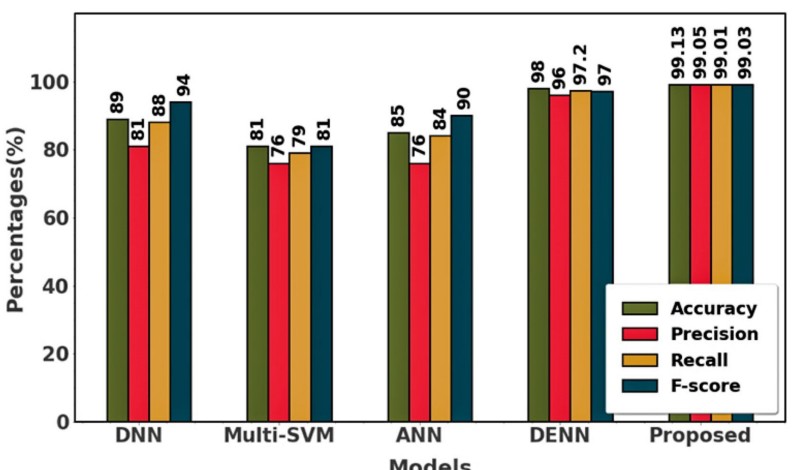

**Figure 14 Differentiation of prior with proposed using dataset 3.**

**Table 4 Comparison of existing approaches with proposed using dataset 3.**

| Approaches | Accuracy (%) | Precision (%) | Recall (%) | F-Score (%) |
|---|---|---|---|---|
| DNN | 89 | 81 | 88 | 94 |
| Multi-SVM | 81 | 76 | 79 | 81 |
| ANN | 85 | 76 | 84 | 90 |
| DENN | 98 | 96 | 97.2 | 97 |
| Proposed | 99.13 | 99.05 | 99.01 | 99.03 |

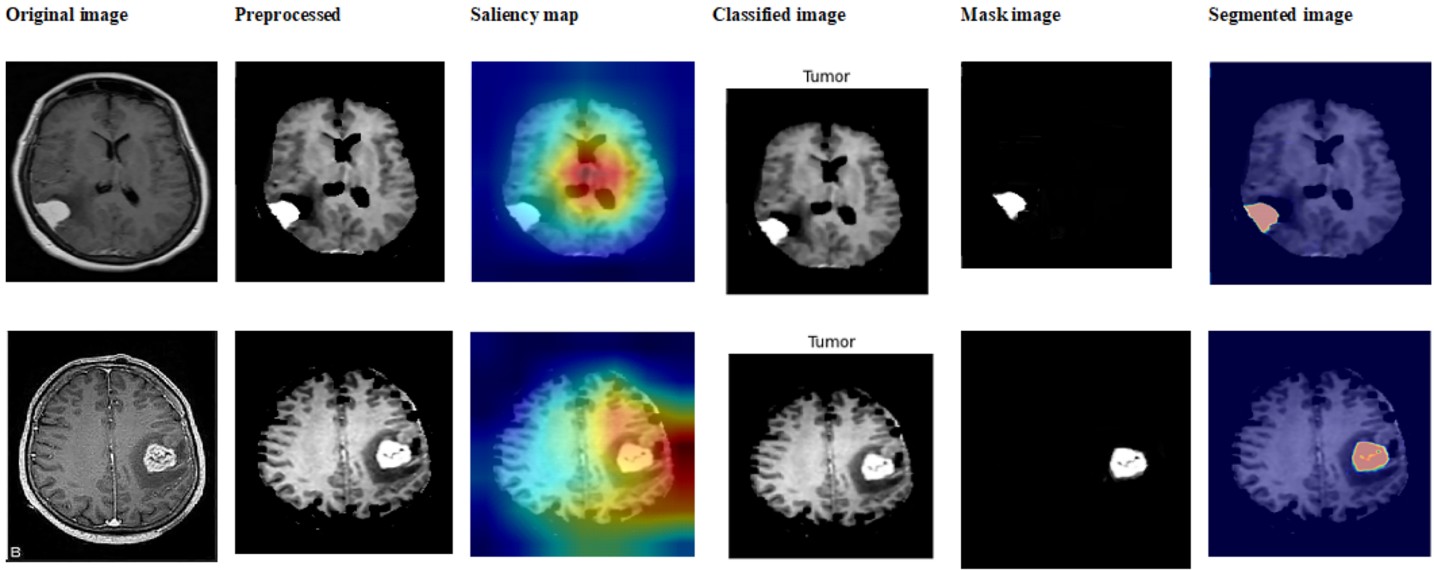

Original image | Preprocessed | Saliency map | Classified image | Mask image | Segmented image

**Figure 15 Sample outputs of dataset 4.**

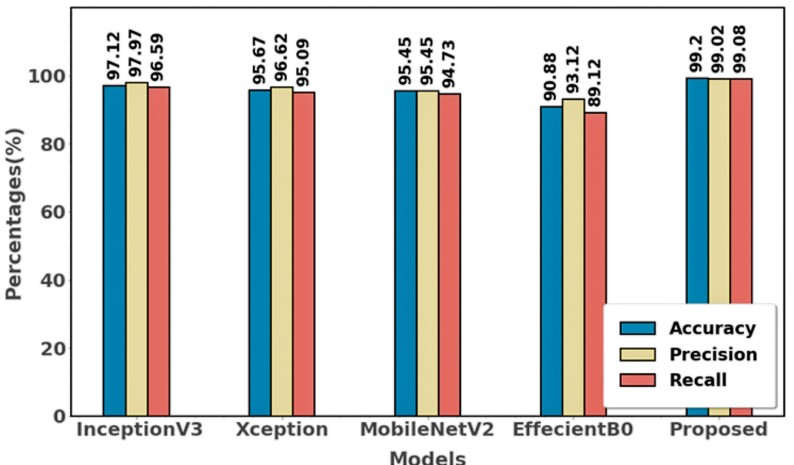

**Figure 16 Differentiation of proposed with an existing method using dataset 4.**

The Table 7 and Fig. 17 presents a comparison of different segmentation strategies based on metrics including weighted IOU, mean BF-Score, mean accuracy, and mean IOU.

With the best results across all metrics, including a mean IOU of 99.31% and a global accuracy of 99.21%, the suggested method beats all other approaches. This illustrates how well it performed correctly segmenting brain tumor images from the MRI dataset. The proposed method outperforms other approaches in all metrics compared to other methods for tumor classification and segmentation, as Table 7 and Fig. 17 illustrate.

It maintains the lowest MAE (0.024), MSE (0.03), and RMSE (0.164) while achieving the highest accuracy (99.42%), recall (99.21%), and precision (99.14%). The proposed approach achieves the highest accuracy of 99.42%, significantly outperforming the next

**Table 5 Comparison of existing approaches with proposed using dataset 4.**

| Approaches | Accuracy (%) | Precision (%) | Recall (%) |
|---|---|---|---|
| InceptionV3 | 97.12 | 97.97 | 96.59 |
| Xception | 95.67 | 96.62 | 95.09 |
| MobileNetV2 | 95.45 | 95.45 | 94.73 |
| EffecientB0 | 90.88 | 93.12 | 89.12 |
| Proposed | 99.20 | 99.02 | 99.08 |

**Table 6 Performance comparison over various image enhancement approaches.**

| | HE | | CLARE | | BBHE | | NSCT | | Proposed method | |
|---|---|---|---|---|---|---|---|---|---|---|
| | PSNR | Contrast | PSNR | Contrast | PSNR | Contrast | PSNR | Contrast | PSNR | Contrast |
| Meningioma | 19.12 | 39.30 | 21.22 | 40.98 | 20.01 | 38.12 | 23.93 | 74.89 | 25.67 | 98.57 |
| Glioma | 20.34 | 27.21 | 20.97 | 29.01 | 19.94 | 28.98 | 24.99 | 71.22 | 28.06 | 96.85 |
| Pituitary | 18.95 | 41.01 | 22.01 | 42.97 | 21.05 | 42.02 | 25.46 | 79.98 | 32.14 | 99.01 |

**Table 7 Differentiation over existing with proposed segmentation approaches.**

| Approaches | Global accuracy | Mean IOU | Mean accuracy | Mean BF-Score | Weighted IOU |
|---|---|---|---|---|---|
| U- SegNet | 98.24 | 64.79 | 91.68 | 84.51 | 98.22 |
| Seg-UNet | 99.11 | 73.40 | 93.12 | 85.07 | 98.63 |
| U-Net | 98.08 | 59.21 | 90.42 | 63.49 | 97.56 |
| SegNet3 | 97.62 | 53.64 | 89.32 | 77.26 | 95.85 |
| Res-SegNet | 98.85 | 68.91 | 93.35 | 82.14 | 98.29 |
| SegNet5 | 98.19 | 60.21 | 91.78 | 64.46 | 98.56 |
| Proposed | 99.21 | 99.31 | 98.99 | 99.24 | 99.35 |

best method, TA-DCAE, which has an accuracy of 97.28%. This shows that the proposed method correctly classifies a larger proportion of brain tumor cases. Furthermore, the model's recall of 99.21% demonstrates its exceptional ability to detect positive cases, while the precision of 99.14% indicates that nearly all identified tumors are correctly predicted. Furthermore, the proposed method has the lowest MAE of 0.024 and RMSE of 0.164, indicating its consistency in making accurate predictions with minimal error. These results outperform previous methods, such as Dense Conv AE and Con-AE, which have higher MAE and RMSE values. This illustrates how the suggested method outperforms other approaches regarding performance and robustness. Overall classification differentiation is shown in Fig. 18 and Table 8.

Figure 19 compares the effectiveness of existing and proposed approaches for various k-fold values. The k-fold validation evaluation examines the categorizing efficiency of the data that is new. If the proposed framework performs well in k-fold examination, it will be better suited for new data and real-time brain tumor predictions. This study examines

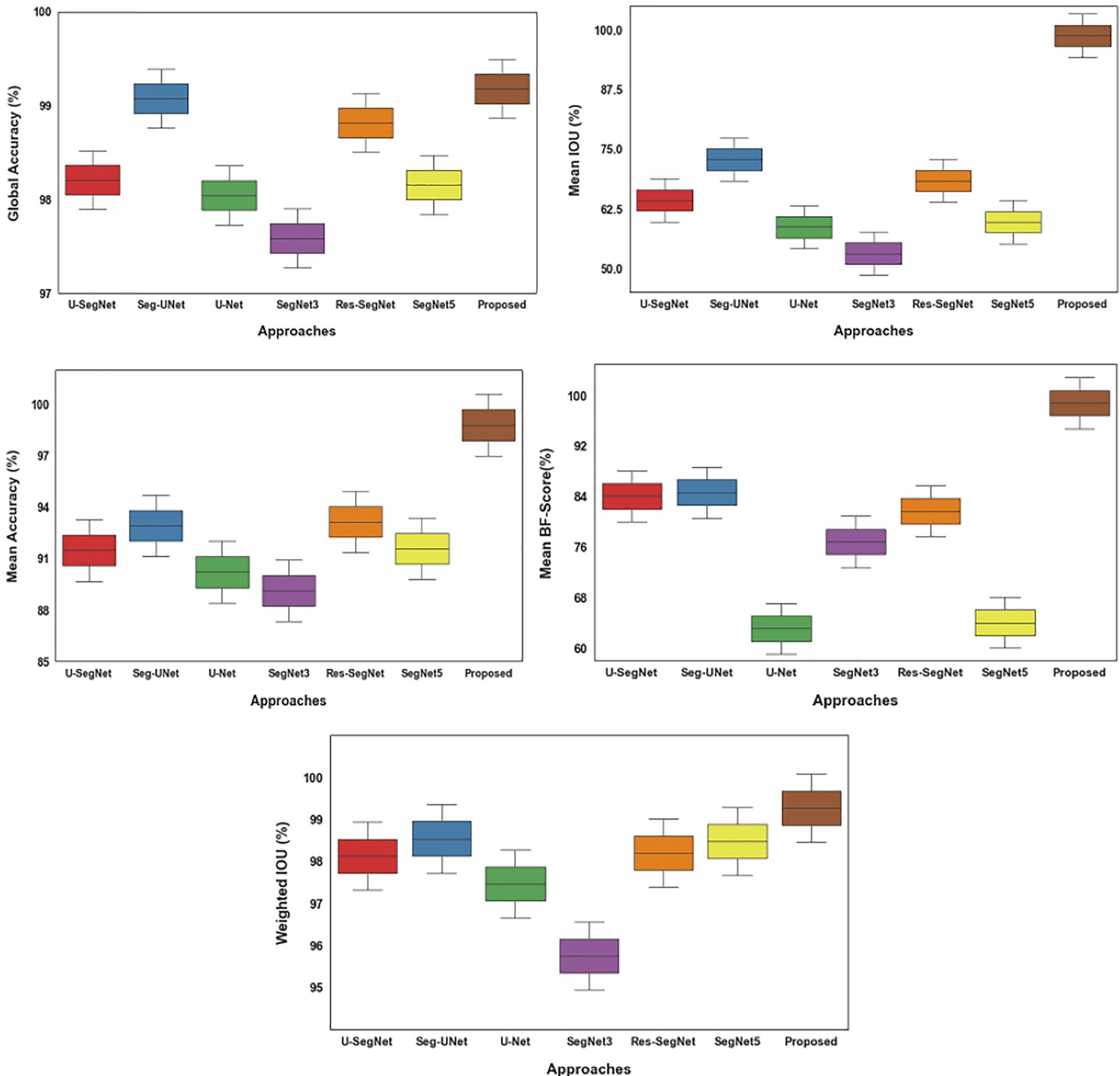

**Figure 17 Differentiation of existing with proposed segmentation approaches.**

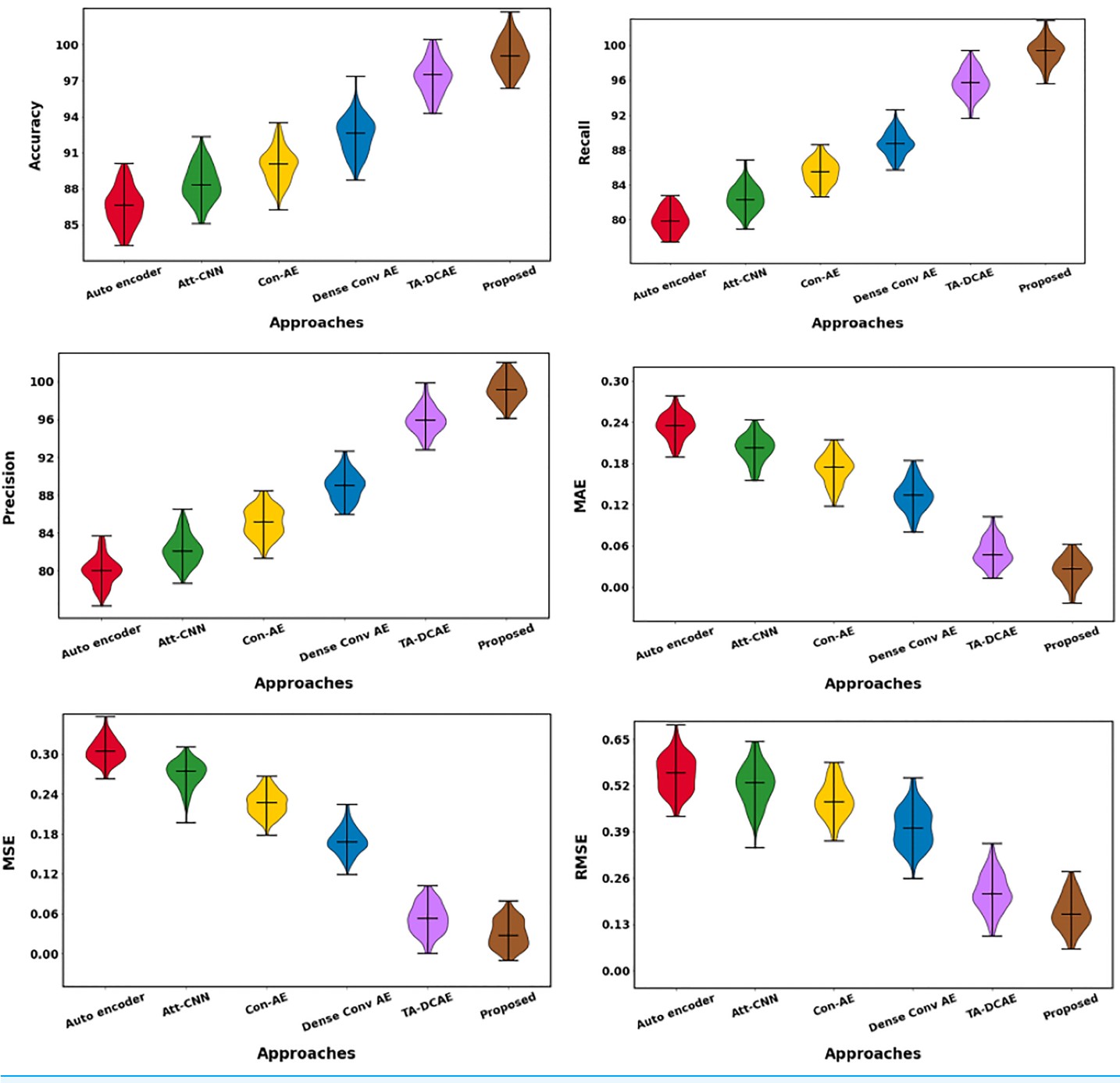

**Figure 18 Overall classification evaluation.**

brain tumor prediction algorithms at k-fold values of 5, 10, 15, 20, and 25. Performance indicators include accuracy, kappa, recall, specificity, F1 score, and precision.

Table 9 shows the differentiation between the existing approach and the proposed approach. While differentiating from existing methods, the proposed approach yields a greater solution.

Table 8 Overall comparison of existing approaches.

| Approaches | Accuracy | Recall | Precision | MAE | MAP | RMSE |
| --- | --- | --- | --- | --- | --- | --- |
| Autoencoder | 86.6 | 79.9 | 80 | 0.235 | 0.303 | 0.551 |
| Att-CNN | 88.24 | 82.35 | 82.41 | 0.207 | 0.269 | 0.519 |
| Con-AE | 90.09 | 85.13 | 85.18 | 0.174 | 0.227 | 0.476 |
| Dense Conv AE | 92.48 | 88.73 | 88.87 | 0.132 | 0.171 | 0.414 |
| TA-DCAE | 97.28 | 95.92 | 95.94 | 0.046 | 0.05 | 0.225 |
| Proposed | 99.42 | 99.21 | 99.14 | 0.024 | 0.03 | 0.164 |

Figure 20 and Table 10 show the computational complexity analysis proposed with existing approaches. While differentiating from existing approaches, the proposed approach takes less time to execute, which is 0.11 s.

The knowledge gap in this study centers around the challenges in accurately classifying brain tumors from MRI scans due to the wide variability in tumor size, shape, and location. Existing methods are often inadequate in capturing these complexities, leading to less accurate classifications and slower processing times. Additionally, many current approaches struggle to handle class imbalance in datasets, where certain tumor types are underrepresented, and further affecting classification accuracy.

Our study contributes to closing this gap by introducing a novel deep learning-based approach, ERSACA-Net, which combines an Extension residual structure with an Adaptive Channel Attention Mechanism to enhance classification accuracy. We also employ Enhanced Res2Net for multi-scale feature extraction, capturing essential details of tumors, and the Binary Chaotic Transient Search Optimization (BCTSO) Algorithm for selecting the most relevant features, thereby reducing computational complexity. Moreover, we address class imbalance using the Conditional Tabular Generative Adversarial Network (CTGAN), ensuring the model performs effectively across all tumor types. This integrated approach not only improves accuracy and processing time but also offers a more robust solution to the challenges present in previous methods.

Table 11 displays the ablation study of the proposed framework across various modules. The algorithm is evaluated without considering pre-processing, extraction of features, and segmentation. Applying pre-processing, feature extraction, and segmentation improves the efficiency of the model that was proposed. The proposed model requires all processes to achieve accurate brain tumor categorization.

## Discussion on performance of proposed method

Compared to current methods, the proposed pipeline's results show notable improvements in a number of areas when it comes to classifying meningioma *vs* pituitary tumors. The performance that was attained according to each parameter is broken down here:

- With an accuracy value of 99.42, the method demonstrates high precision in correctly identifying tumors without misclassifying tumors.

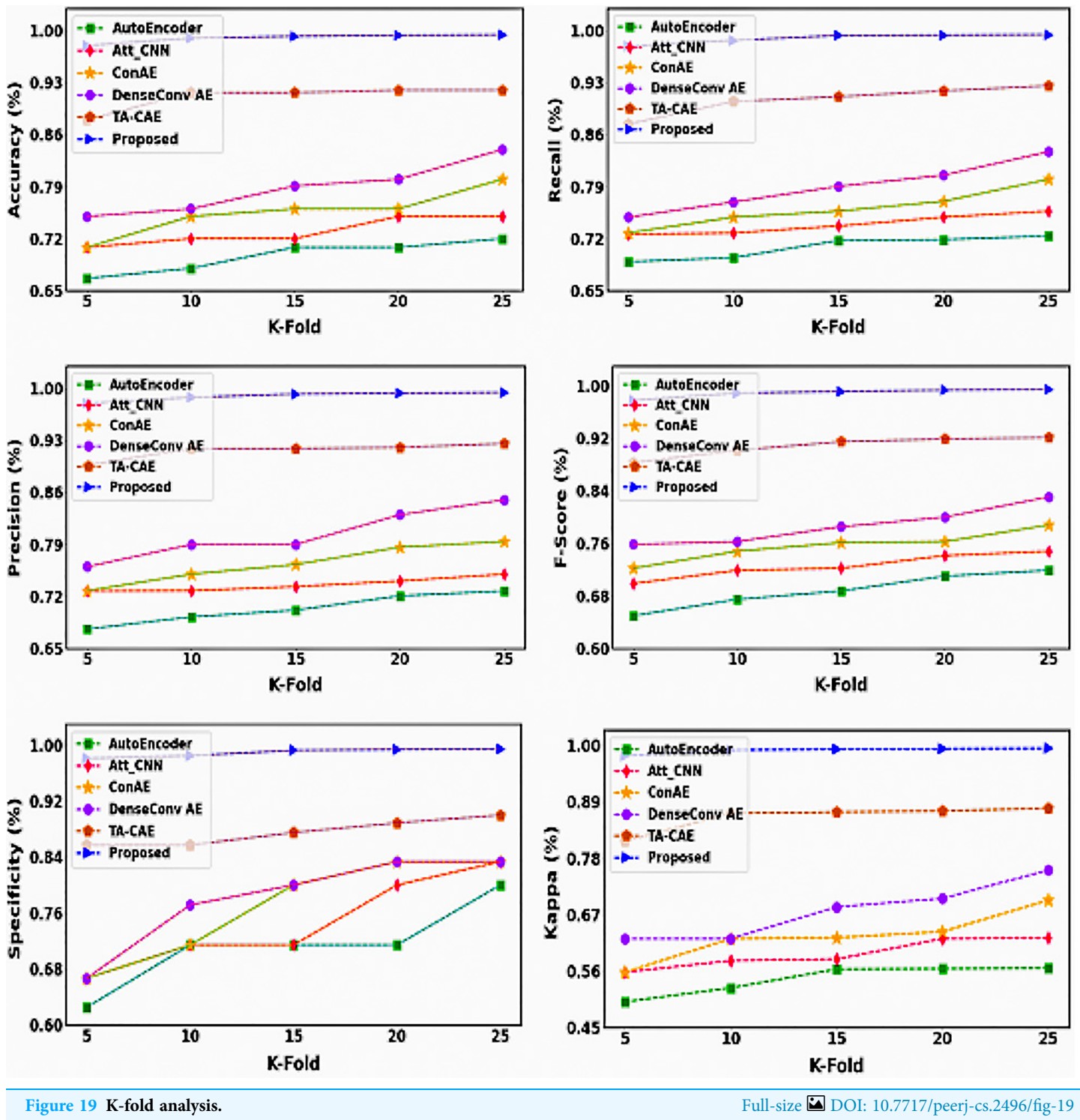

**Figure 19 K-fold analysis.**

**Table 9 Comparison of existing related work with proposed.**

| Reference | Method used | Accuracy (%) | Precision (%) | Recall (%) |
|---|---|---|---|---|
| *Kulshreshtha & Nagpal (2024)* | CNN | 98.23 | 95.14 | – |
| *Arumugam et al. (2024)* | CSA-MLP | 98.56 | 96.52 | – |
| *Mandle, Sahu & Gupta (2024)* | WSSOA | 99.29 | 99.04 | 98.79 |
| *Lee, Chae & Cho (2024)* | Patterned GridMask | 99.74 | – | – |
| *Alagarsamy, Govindaraj & Senthilkumar (2023)* | IT2FLS-ABC | 99.01 | – | – |
| *Asiri et al. (2023)* | CNN | 98.45 | – | – |
| *Rao & Karunakara (2022)* | KSVM | 98.26 | 95.41 | – |
| *Sekhar et al. (2021)* | SVM | 97.56 | 94.65 | 96.66 |
| *Amin et al. (2024)* | RF | 98.87 | – | – |
| Proposed | | 99.42 | 99.14 | 99.21 |

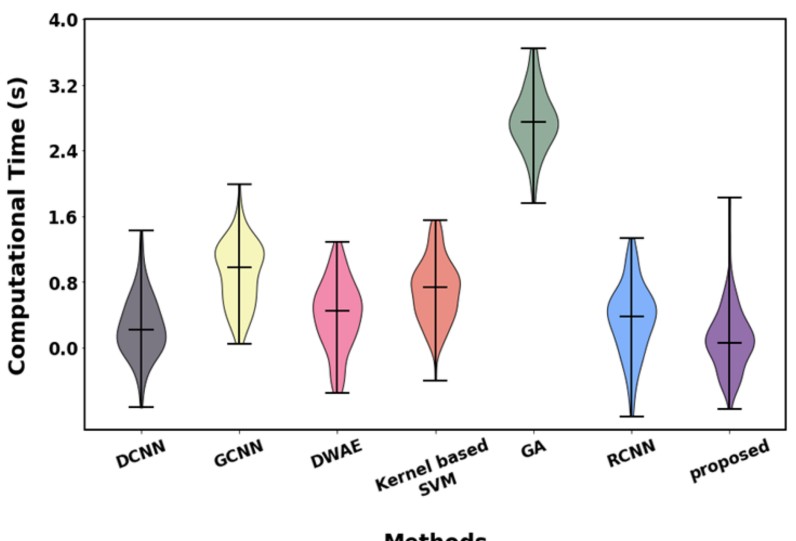

**Figure 20 Graphical illustration of computational analysis.**

**Table 10 Computational complexity.**

| Method | Computational time (s) |
|---|---|
| DCNN | 0.23 |
| GCNN | 0.92 |
| DWAE | 0.43 |
| Kernel-based SVM | 0.83 |
| GA | 2.8 |
| RCNN | 0.32 |
| Proposed | 0.11 |

**Table 11 Ablation study.**

| Metrics | Without preprocessing | Without feature extraction | Without segmentation | Proposed |
|---|---|---|---|---|
| Accuracy | 95.66 | 94.12 | 92.45 | 99.42 |
| Recall | 94.87 | 89.54 | 91.45 | 99.21 |
| Precision | 95.60 | 92.74 | 93.88 | 99.14 |
| F-Measure | 93.41 | 90.14 | 94.17 | 99.175 |
| MAP | 0.0821 | 0.1152 | 0.1547 | 0.03 |
| MAE | 0.0947 | 0.1348 | 0.1147 | 0.024 |
| RMSE | 0.2651 | 0.3241 | 0.4717 | 0.164 |

- A precision of 99.14 indicates that pituitary tumors and meningioma tumors can be distinguished with this method's efficacy.
- Notably, the suggested approach shows efficiency and computational speed with an execution time of 0.11 s less than previous works.

The findings suggest that the suggested approach outperforms current accuracy, recall, precision, MAE, RMSE and IOU metrics. These advancements in medical imaging have a lot of potential to produce more accurate and effective diagnoses. This can ultimately improve the care of patients with pituitary tumors and meningioma by speeding up treatment decisions.

The proposed ERSACA-Net effectively classifies and segments brain tumors by leveraging adaptive residual and channel attention mechanisms, addressing variability in tumor size, shape, and location. Compared to existing deep learning methods, ERSACA-Net enhances feature learning, leading to improved classification outcomes. The use of Enhanced Res2Net for feature extraction and BCTSO for feature selection significantly reduces computational complexity while maintaining high classification accuracy. The novel LWIFCM_CSA approach and CTGAN effectively handle class imbalance and segmentation, providing more reliable segmentations and model robustness. This approach is highly applicable in the medical field, offering automated brain tumor analysis that reduces medical professionals' workload and provides more consistent assessments, ultimately improving patient outcomes. Regarding explainability, the proposed method's adaptive channel attention mechanisms identify important features influencing classification, allowing visualization of attention maps. This enhances transparency, making it easier for medical professionals to understand and trust the model's decisions, thus intertwining explainability with the proposed approach.

## Proposed strengths and limitations: a comprehensive analysis
The approach we proposed has the advantages listed below, along with some disadvantages.

- One notable aspect of the research is the implementation of an advanced automated segmentation pipeline. This new method streamlines the segmentation and categorizing of brain tumors in MRI images, potentially improving accuracy. It incorporates sophisticated preprocessing techniques like diffusion filtering and contrast-limited adaptive histogram equalization (ACCLAHE).

- A significant advantage is the segmentation process's use of the Novel LWIFCM_CSA algorithm. Acknowledged for its effectiveness in defining structures in medical images, it enhances the precision of identifying anomalous areas associated with brain tumors. However, care must be taken because LWIFCM is sensitive to noise and outliers. Hybrid strategies were used to overcome this, combining FCM with additional techniques.

- By adding an ERSACA-Net classifier, the suggested method can now handle more complex classification tasks.

- The approach has a noticeably faster processing time (0.11 s), which is an important feature for real-world application in clinical settings where prompt diagnosis is essential. This efficiency is a noteworthy strength that adds a new dimension to the suggested strategy when compared to traditional methods. Although the proposed MRI-based automatic tumor segmentation and categorization method has yielded encouraging results, it is crucial to recognize its inherent limitations:

Absence of clinical validation: The suggested approach has not been validated in a large-scale clinical context, which could provide a risk to its safety and dependability in practical situations.

• Future data drift and updates: As imaging technology advances, clinical procedures alter, or new datasets become available, the method's effectiveness may erode over time. To stay relevant, medical imaging must be updated frequently and adjusted to new developments.

• Limited generalization: There may be difficulties translating the method's optimal performance on the particular CE-MRI database used for testing to other datasets or populations. These issues will be resolved in subsequent iterations.

• Dependency on image quality: Artifacts, low resolution, or other quality problems in the input MRI images could cause segmentation and classification accuracy to be compromised, thereby compromising the effectiveness of the suggested method. Preprocessing must be validated on large databases in order to improve performance, and this is a critical step in improving future brain tumor techniques.

The proposed approach's robustness and practicality will be greatly strengthened by addressing these limitations through future research, especially in a variety of clinical settings.

## Limitation and future scope

One limitation is that the model's performance may vary when applied to different imaging modalities or datasets not included in our experiments, potentially affecting its robustness across diverse clinical environments. Additionally, while our approach efficiently handles class imbalance and enhances classification accuracy, its reliance on specific MRI features may limit generalizability to other types of medical images or unseen conditions. To address these concerns, future work will explore the adaptability of the method to other datasets and imaging types to ensure broader applicability.

## CONCLUSION

This work provides a thorough pipeline for brain tumor automated segmentation and categorization. The suggested approach is proof of creativity and effectiveness in the identification of brain tumors. Our method emphasizes its innovative nature by introducing distinct modifications at each step. Its actual distinctiveness comes from the careful blending and modification of these methods, which are adapted to the nuances of magnetic resonance (MR) images. We have acknowledged and addressed the challenges posed by MR images and have deliberately adjusted established methodologies by carefully choosing and integrating them to maximize their performance. Significantly, our approach presents a new framework that includes cutting-edge preprocessing methods such as adaptively clipped contrast-limited adaptive histogram equalization (ACCLAHE). To tackle the class imbalance problem CTGAN is introduced. Brain tumor essential features are extracted and selected based on Enhanced Res2Net and BCTSO algorithm. Glioma, meningioma and pituitary tumors are categorized by using Deep learning based novel ERSACA-Net model. Finally to segment the affected tumors we ensemble local-information weighted intuitionistic fuzzy C-means clustering algorithm (LWIFCM) and Chameleon Swarm Algorithm (CSA).

To investigate and analyze the proposed approach performance we utilized four benchmark datasets. While differentiating with existing approaches, our proposed approach gain superior performances in 99.42% accuracy, 99.21% precision, 99.14% recall, 0.024% MAE, 0.164% RMSE and 0.03% MSE. This shows while comparing with existing state of the art approaches our proposed approach gain superior performances.

## ABBREVIATION

| | |
|---|---|
| **ACCLAHE** | Adaptively Clipped contrast-limited adaptive histogram equalization |
| **LWIFCM** | Local-information weighted intuitionistic Fuzzy C-means clustering algorithm |
| **CSA** | Chameleon Swarm Algorithm |
| **MRI** | Magnetic Resonance Imaging |
| **ERSACA-Net** | Extension residual structure and Adaptive Channel Attention Mechanism |
| **CTGAN** | Conditional Tabular Generative Adversarial Network |
| **BCTSO** | Binary Chaotic Transient Search Optimization |
| **IFAS** | Enhanced fully automatic segmentation |
| **CSA-MLP** | Crossover Smell Agent Optimized Multilayer Perception |

| CNN | Convolutional Neural Network |
|---|---|
| DCNN | Deep Convolutional Neural Network |
| WSSOA | Whale Social Spider-based Optimization Algorithm |
| IT2FLS | Interval Type-II fuzzy logic system |
| SVM | Support vector machine |
| HHO | Harris Hawks Optimization |
| SSD | Social Ski Driver |
| LBP | Local Binary Pattern |
| GWF | Gabor wavelet features |
| SFTA | Segmentation-Based Fractal Texture Analysis |

### Funding

The authors received no funding for this work.

### Competing Interests

The authors declare that they have no competing interests.

### Author Contributions

- Nadenlla RajamohanReddy conceived and designed the experiments, performed the experiments, prepared figures and/or tables, authored or reviewed drafts of the article, and approved the final draft.
- G. Muneeswari analyzed the data, performed the computation work, prepared figures and/or tables, authored or reviewed drafts of the article, and approved the final draft.

### Data Availability

The Brain Tumors Segmentation dataset is available at figshare: RajamohanReddy, Nadenlla (2024). Brain Tumors Segmentation. figshare. Dataset. https://doi.org/10.6084/m9.figshare.26037178.v2.

The Brain Tumor-MRI dataset is available at Kaggle: https://www.kaggle.com/datasets/masoudnickparvar/brain-tumor-mri-dataset.

The Brats 2020 dataset is available at Kaggle: https://www.kaggle.com/datasets/awsaf49/brats2020-training-data.

The Brain Tumor Classification is available at Kaggle: https://www.kaggle.com/datasets/rahimanshu/figshare-brain-tumor-classification.

The Br35H dataset is available at Kaggle: https://www.kaggle.com/datasets/ahmedhamada0/brain-tumor-detection.

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
