# Peer review of "Advancing multi-categorization and segmentation in brain tumors using novel efficient deep learning approaches"

_PeerJ Computer Science, doi:10.7717/peerj-cs.2496_

## Round 0.1 · original submission · Major Revisions

Dear authors,
You are advised to critically respond to all comments point by point when preparing a new version of the manuscript and while preparing for the rebuttal letter. Please address all the comments/suggestions provided by the reviewers.

Kind regards,
PCoelho

·

Basic reporting

The article under consideration seems interesting and emerging. Furthermore, the authors have justified the study by demonstrating the results by using four datasets. The study, A Hybrid Deep Learning-Based Approach for Brain Tumor Classification can also be considered for further improving the literature section.

Experimental design

The demonstrated experimental design was state-of-the-art. I have no further comments on it.

Validity of the findings

The demonstrated results are good and scientifically sound.

·

Basic reporting

It is necessary to improve the extremely poor English level. The introduction and background information in the article should be adequate to show how the work fits into the larger body of knowledge. The paper needs to be revised and improved in terms of its structure.

Experimental design

The knowledge gap under investigation should be identified, and the study's contribution to closing that gap should be explained.

The manuscript does not clearly outline the criteria for selecting the datasets used in the experiments. It would be beneficial to provide a more detailed justification for the choice of datasets and how they represent the problem space.
There is a lack of detailed description regarding the pre-processing steps applied to the data. Understanding these steps is crucial for assessing the reproducibility and reliability of the results.
The comparison with other state-of-the-art methods is not comprehensive. The authors should include more benchmarks and a discussion on why certain methods were chosen for comparison.

Validity of the findings

The performance metrics reported in the manuscript are promising, but the statistical significance of these results is not discussed. Including confidence intervals or statistical tests would strengthen the validity of the findings.
The manuscript lacks an ablation study to demonstrate the contribution of each component of the proposed approach. Such a study would help in understanding the individual impact of each component on the overall performance.
There is a need for a more thorough analysis of potential limitations and how they might affect the generalizability of the proposed method.
I hope these additional details help in understanding my recommendation for major revisions. Please let me know if further elaboration on any specific point is required.

·

Basic reporting

1. The authors do not explicitly mention in the introduction, what are the challenges with the existing approaches, which motivated them to propose a novel approach.
2. The motivation of the paper seems too generic. In the paper, the authors didn’t explicitly explain the motivation of the proposed approach. What is different about the proposed approach than the previous research approaches? It should be described in the introduction. That should be the main driver for the proposed approach. Currently, I haven’t configured what the proposed approach tries to achieve.
3. Related work is badly organized, too many short paragraphs. It should be divided into different sub-sections based on similar work. For example ML-based approaches, Deep learning-based approaches, etc.
4. A comparative study with the existing approaches will elaborate further on the point of concentration of the proposed approach

Experimental design

5. Authors mentioned the feature extraction from the MRI images. While deep/transfer learning algorithms are good in extracting features from the images, it is, therefore, needs to elaborate on why the author is required to apply approaches to extract images from the MRI.
6. The authors didn’t mention how many features were identified.
7. The paper needs thorough revision, there are many grammatical mistakes in the paper
8. Why the dataset text is highlighted?
9. Random data splitting at this level is not acceptable, the author is advised to use kfold training and validation approach, which is a more reliable and recommended method.
10. The software and hardware running Environment in this study should be added.
11. It is not clear whether the proposed approach, compared with the state-of-the-art, utilized the same dataset or a different one; if different, it will be better to add some research approaches that use similar datasets for evaluation.
12. Research questions are missing in the paper, which is an essential part of the scientific paper.
13. The discussion section should be extended by adding more insightful information. The research questions added in the paper should be answered in the discussion section. It should always elaborate on the application of the proposed approach. Also, the authors should discuss about explainability of the proposed approach, as it is considered a hot area, how explainability can be intertwined with the proposed approach?
14. Did the authors perform an ablation study to compare with different models?
15. When re-arranging the related and introduction work, the below references may be beneficial to compare your proposed work, improving the paper by adding the latest references
A Hybrid Deep Learning-Based Approach for Brain Tumor Classification.
A Robust End-to-End Deep Learning-Based Approach for Effective and Reliable BTD Using MR Images
Enhancing explainability in brain tumor detection: A novel DeepEBTDNet model with LIME on MRI images

Validity of the findings

Future work is not discussed in the paper.

Reviewer 4 ·

Basic reporting

The paper is well-structured with clear articulation of the research problem and objectives. It provides a comprehensive description of the methods and introduces novel techniques. However, there are minor issues with grammar and flow that could be improved for better readability.

Experimental design

The experimental design is robust, leveraging four benchmark datasets to validate the proposed method. The integration of novel algorithms and preprocessing techniques demonstrates a well-thought-out approach. However, additional details on the dataset selection criteria and hyperparameter tuning could enhance reproducibility and clarity.

Validity of the findings

The findings are supported by strong performance metrics, showing superior accuracy, precision, recall, and error rates compared to existing approaches. The use of multiple datasets adds credibility to the results. However, further validation on more diverse datasets and a comparison with a broader range of state-of-the-art methods would strengthen the generalizability of the conclusions.

---

## Round 0.2 · accepted · Accept

Dear authors, we are pleased to verify that you meet the reviewer's valuable feedback to improve your research.

Although in the last round there was only feedback from one of the previous reviewers, they were one of the reviewers who had more critical comments, and after my own verification, the manuscript is ready to be accepted.

Thank you for considering PeerJ Computer Science and submitting your work.

·

Basic reporting

The authors have improved the paper by incorporating all my previous comments in the paper. Therefore, the paper is accepted in its current form.

Experimental design

The experiment part of the paper has been improved by incorporating the changes.

Validity of the findings

The approach has been additionally validated with Kfold and authors performed the ablation study,